# The ability of macroalgae to mitigate the negative effects of ocean acidification on four species of North Atlantic bivalve

Craig S. Young[1] and Christopher J. Gobler[1]

[1]Stony Brook University, School of Marine and Atmospheric Sciences, Southampton, NY 11968, USA
*Correspondence to:* Christopher J. Gobler (Christopher.gobler@stonybrook.edu)

**Abstract.** Coastal ecosystems can experience acidification via upwelling, eutrophication, riverine discharge, and climate
change. While the resulting increases in $p\text{CO}_2$ can have deleterious effects on calcifying animals, this change in carbonate chemistry may benefit some marine autotrophs. Here, we report on experiments performed with North Atlantic populations of hard clams (*Mercenaria mercenaria*), eastern oysters (*Crassostrea virginica*), bay scallops (*Argopecten irradians*), and blue mussels (*Mytilus edulis*) grown with and without North Atlantic populations of the green macroalgae, *Ulva*. In 6 of 7 experiments, exposure to elevated $p\text{CO}_2$ levels (~1,700 µatm) resulted in depressed shell- and/or tissue-based growth rates of
bivalves compared to control conditions whereas rates were significantly higher in the presence of *Ulva* in all experiments. In many cases, the co-exposure to elevated $p\text{CO}_2$ levels and *Ulva* had an antagonistic effect on bivalve growth rates whereby the presence of *Ulva* under elevated $p\text{CO}_2$ levels significantly improved their performance compared to the acidification only treatment. Saturation states for calcium carbonate ($\Omega$) were significantly higher in the presence of *Ulva* under both ambient and elevated $\text{CO}_2$ delivery rates and growth rates of bivalves were significantly correlated with $\Omega$ in six of seven
experiments. Collectively, the results suggest that photosynthesis and/or nitrate assimilation by *Ulva* increased alkalinity, fostering a carbonate chemistry regime more suitable for optimal growth of calcifying bivalves. This suggests that large natural and/or aquacultured collections of macroalgae in acidified environments could serve as a refuge for calcifying animals that may otherwise be negatively impacted by elevated $p\text{CO}_2$ levels and depressed $\Omega$.

## 1 Introduction

The continued delivery of $\text{CO}_2$ into surface oceans is expected to cause significant shifts in pools of inorganic carbon by the end of this century, with projected increases in $\text{CO}_2$ and $\text{HCO}_3^-$ and decreases in $\text{CO}_3^{2-}$ and the saturation states of calcite ($\Omega_{\text{calcite}}$) and aragonite ($\Omega_{\text{aragonite}}$) (Feely et al., 2009; Meehl et al., 2007). Beyond the delivery of $\text{CO}_2$ via the combustion of fossil fuels, upwelling, riverine discharge, eutrophication-accelerated microbial respiration all represent strong
sources of $\text{CO}_2$ into coastal zones (Cai et al., 2011; Feely et al., 2008; Melzner et al., 2013; Salisbury et al., 2008; Wallace et al., 2014). Eutrophication-enhanced respiration in coastal zones can lead to the accumulation of respiratory $\text{CO}_2$ that can

exceed concentrations projected for the end of the century (>2,000 µatm), as well as result in the undersaturation of aragonite ($\Omega_{aragonite} < 1$; Cai et al., 2017; Wallace et al., 2014).

Calcifying organisms are highly vulnerable to the projected shifts in the various pools of total dissolved inorganic carbon (DIC), with the deleterious effects of ocean acidification being well-documented for corals (Hoegh-Guldberg et al., 2007; Kleypas et al., 1999), coralline algae (Gao and Zheng, 2010; Martin and Gattuso, 2009), and bivalves (Barton et al., 2012; Gazeau et al., 2007; Talmage and Gobler, 2011). Acidification-induced reductions in $\Omega_{calcite}$ and $\Omega_{aragonite}$ can result in lowered survivorship and inhibited growth for larvae and juvenile stage bivalves (Gobler et al., 2014; Green et al., 2009; Talmage and Gobler, 2011; Waldbusser et al., 2015a). Since bivalves provide numerous ecosystem and economic services (Newell, 2004), and elevated $p$CO$_2$ is a common occurrence in many coastal ecosystems (Feely et al., 2008; Salisbury et al., 2008; Wallace et al., 2014), it is important to understand how other co-occurring estuarine life will respond to high $p$CO$_2$ conditions and may, in turn, effect acidification-vulnerable organisms such as bivalves.

Contrary to the negative effects of increased CO$_2$ on calcifying organisms, previous studies have shown that some photosynthetic organisms, such as seagrasses (Koch et al., 2013; Palacios and Zimmerman, 2007), phytoplankton (Fu et al., 2012; Hattenrath-Lehmann et al., 2015), and macroalgae (Olischläger et al., 2013; Young and Gobler, 2016) may benefit from a high CO$_2$ environment. Such photosynthetic autotrophs may also have the capacity to buffer carbonate chemistry, potentially alleviating the harmful effects of excessive CO$_2$ on calcifying organisms. For example, prior studies have observed that daytime productivity within seagrass meadows can increase pH and $\Omega_{aragonite}$ which, under future acidified conditions, may provide temporal refuge for calcifying animals (Garrard et al., 2014; Hendriks et al., 2014). Given the significant global declines in seagrass (Orth et al., 2006; Short et al., 2011; Waycott et al., 2009), as well as the overgrowth of seagrass beds by macroalgae (McGlathery, 2001; Valiela et al., 1997), it is plausible macroalgae may more commonly provide similar ecosystem services. While future increases in CO$_2$ may promote the growth of fast-growing, macroalgae such as *Ulva* (Björk et al., 1993; Olischläger et al., 2013; Young and Gobler, 2016, 2017) and could, in turn, could provide chemical resilience for calcifying organisms in acidified environments (Anthony et al., 2013; Wahl et al., 2017), such interactions have yet to be fully explored.

Recent studies have demonstrated that populations of *Ulva rigida* from Northwest Atlantic coastal waters experience enhanced growth under elevated CO$_2$ concentrations (Young and Gobler, 2016, 2017). While past studies have suggested that macroalgae may buffer carbonate chemistry to the benefit of bivalves (Anthony et al., 2013; Wahl et al., 2017), no study has assessed how *Ulva*, a common macroalga known to undergo enhanced growth under acidified and eutrophic conditions, may affect bivalves under CO$_2$-enhanced conditions. The objective of this study, therefore, was to assess how elevated $p$CO$_2$ and the presence of *Ulva* influences the growth and survival of seven cohorts of juvenile bivalves indigenous to North Atlantic, including hard clams (= northern quahogs; *Mercenaria mercenaria*), eastern oysters (*Crassostrea virginica*), bay scallops (*Argopecten irradians*), and blue mussels (*Mytilus edulis*). Small- and large-sized individuals of bivalves were assessed for three species given the effects of ocean acidification can be size- and species-dependent for juvenile bivalves (Talmage and Gobler, 2011; Waldbusser et al., 2010). Each bivalve cohort was grown with

and without elevated $CO_2$ levels as well as with and without *Ulva*. Growth and survival of the bivalves were quantified along with carbonate chemistry within experimental vessels.

## 2 Methods

### 2.1 Experimental design

Seven experiments were performed to assess the effects of elevated $pCO_2$ and the presence of *Ulva* on the growth and survival of *M. mercenaria*, *C. virginica*, *A. irradians*, and *M. edulis*. Experiments using smaller bivalves (1 – 5 mm) were performed in 1 L polycarbonate vessels, while experiments with larger bivalves (20 – 21 mm) were performed in larger, 8 L polycarbonate vessels. All containers were acid washed (10% HCl) and liberally rinsed with deionized water prior to use. The experimental vessels were placed in an environmental control chamber set to a consistent temperature (~21°C), light intensity (~200 µmol photons $m^{-2}$ $s^{-1}$) and duration (14 h: 10 h light:dark cycle). The light intensity and photoperiod were set to mimic conditions observed at the *Ulva* collection sites during the time of collection (see below). Containers were filled with filtered (0.2µm polysulfone filter capsule, Pall[©]) seawater and were randomly assigned, in quadruplicate, to one of four treatments: a control with ambient $CO_2$ concentrations (~400 µatm) without *Ulva*, a treatment with ambient $CO_2$ levels that received *Ulva*, a treatment with elevated $CO_2$ concentrations (~1700 µatm) without *Ulva*, and a treatment with elevated $CO_2$ and *Ulva*, resulting in 16 experimental containers. Two additional containers were filled with filtered seawater and bubbled in a manner identical to the ambient or elevated $CO_2$ treatments (described below) and were used to obtain initial dissolved inorganic carbon measurements. Continuous dissolved oxygen (DO) measurements were made using HOBO optical DO sensors (Onset[©]) in additional parallel vessels with and without *Ulva* added at the same levels used in experimental vessels and bubbled identically to experimental vessels. All experimental containers for each experiment received nutrient additions (50µM nitrate, 3 µM phosphate) at the beginning of the experiment, as well as after each twice weekly water changes (details below) to ensure nutrient replete growth of *Ulva*. The nutrient and $CO_2$ concentrations used during experiments were within the range of concentrations present in US East Coast estuaries (Baumann and Smith, 2017; Baumann et al., 2015; Wallace et al., 2014; Wallace and Gobler, 2015), and were used during prior experiments that involved *Ulva* from Shinnecock Bay, NY, USA (Young and Gobler, 2016, 2017). Across all experiments, bivalves were fed a mixture of *Isochrysis galbana* and *Chaetoceros muelleri* at rate known to be *ad libitum* (4 x $10^4$ cells $mL^{-1}$ $d^{-1}$; Helm et al., 2004). Microalgal cultures were maintained in exponential phase growth in f/2 media using standard culturing conditions (Helm et al., 2004).

To deliver dissolved gases, each experimental vessel was aerated via a 3.8 x 1.3 cm air diffuser (Pentair) connected to a 1 mL, polystyrene serological pipette inserted to the bottom of each vessel and connected via Tygon tubing to an air source. Containers were subjected to ambient (~400 µatm) and elevated (~1700 µatm) $CO_2$ concentrations via a gas proportionator system (Cole Parmer® Flowmeter system, multitube frame) that mixed ambient air with 5% $CO_2$ gas (Talmage and Gobler, 2010). Gases were mixed and delivered at a flow rate of 2500 ± 5 mL $min^{-1}$ through gang valves into the serological pipettes that fit through an opening in the plexiglass used to cover the experimental containers, turning over

the volume of the experimental containers >1000 times daily. Bubbling began two-to-three days prior to the start of each experiment to allow $CO_2$ concentrations and carbonate chemistry to reach a state of equilibrium. Experiments persisted for ~two weeks. Measurements of pH within containers were made daily with a Honeywell DuraFET III ion-sensitive field-effect transistor-based (ISFET) solid-state pH sensor (± 0.01 pH unit, total scale), which was calibrated with a seawater pH

standard (Dickson, 1993). Continuous measurements of pH were made using an Orion Star A321 Plus electrode (± 0.001 pH unit, NBS scale) calibrated prior to use using National Institute of Standards and Technology (NIST) traceable standards. Continuous pH sensors were used one at a time in singular experimental vessels during occasional experiments, and data were only downloaded and assessed for trends after experiments had been completed. Measurements of pH made with the DuraFET and Orion Star A321 were compared to measurements made spectrophotometrically using *m*-cresol purple

(Dickson et al., 2007), and were found to be nearly identical and never significantly different. Discrete water samples were collected at the beginning and conclusion of experiments to directly measure DIC within each experimental vessel in each treatment (*n*=4 per treatment). The DIC samples were preserved using a saturated mercuric chloride ($HgCl_2$) solution and stored at ~4°C until analysis. Samples were analyzed by a VINDTA 3D (Versatile INstrument for the Determination of Total inorganic carbon) delivery system coupled with a UIC Inc. coulometer (model CM5017O). During the coulometric analysis,

all carbonate species were converted to $CO_2$ gas by the addition of excess hydrogen to the sample and the evolved $CO_2$ gas was subsequently carried into the titration cell of the coulometer. The gas then reacted quantitatively with an ethanolamine-based reagent to generate hydrogen ions, which were titrated with coulometrically-generated $OH^-$, and $CO_2$ was measured by integrating the total change required to titrate the hydrogen ions (Johnson et al., 1993). Final total alkalinity, $\Omega_{aragonite}$, $\Omega_{calcite}$, $p$$CO_2$, and concentrations of $HCO_3^-$, $CO_3^{2-}$ and $OH^-$ (Tables 1 and S1) were calculated from measured levels of DIC, pH,

temperature, and salinity, as well as the first and second dissociation constants of carbonic acid in seawater (Millero, 2010) using the program CO2SYS (http://cdiac.ornl.gov/ftop/co2sys/). For quality assurance, levels of DIC and pH within certified reference material (provided by Dr. Andrew Dickson of the University of California, San Diego, Scripps Institution of Oceanography; batches 158, 159 = 2044, 2027 µmol DIC kg seawater$^{-1}$, respectively) were measured during analyses of every set of samples. The analysis of samples continued only after complete recovery (99.8 ± 0.2 %) of certified reference

material was attained. Actual mean $p$$CO_2$ and pH values were 350 µatm and 8.00, respectively for ambient conditions, and 1750 µatm and 7.38, respectively, for elevated $CO_2$ conditions, values within the range found seasonally in some estuarine environments (Baumann and Smith, 2017; Baumann et al., 2015; Wallace et al., 2014; Wallace and Gobler, 2015). Two-way ANOVAs and post-hoc tests were used to assess significant differences in carbonate chemistry among experimental vessels with the main treatment effects being $p$$CO_2$ (ambient or elevated) and the presence of *Ulva* within SigmaPlot 11.0.

## 2.2 Assessing the effects of elevated $p$$CO_2$ and *Ulva* on juvenile bivalves

The macroalgae used for this study were collected from Shinnecock Bay, NY, USA, (40.85° N, 72.50° W) during low tide. Permission to access this area and collect macroalgae and *M. edulis* was received from the Southampton Town Trustees, Southampton, NY, USA, who hold jurisdiction over Shinnecock Bay. Large, well-pigmented, robust fronds of

*Ulva* were collected and transported to the Stony Brook Marine Science Center in seawater-filled containers within 15 minutes of collection. Previously, ITS sequencing and microscopy was used to determined that the species of *Ulva* that dominated Shinnecock Bay in summer and fall was *Ulva rigida* (Young and Gobler, 2016, 2017) and microscopic examinations during this study indicated this was the species used in all experiments presented here. We refer to the algae

simply as *Ulva* throughout the study due to the plastic nature of the macroalgal taxonomic nomenclature, as well as the high similarity of ITS sequences among species of *Ulva* (Hofmann et al., 2010; Kirkendale et al., 2013).

Well-pigmented, circular sections of *Ulva* (~3.5 cm and ~7 cm for experiments in small containers and large vessels (described below), respectively, with one disk per container) were cut from the larger thalli with care taken to avoid the outer, potentially reproductive region of the algae (Wallace and Gobler, 2015). The weights of *Ulva* used in experiments

relative to the vessels was consistent with the benthic coverage of *Ulva* in Shinnecock Bay (~8 g m$^{-2}$; Gobler and Young, unpublished benthic trawl data) and other estuarine regions (Liu et al., 2015; Sfriso et al., 2001). Experimental disks of *Ulva* were extensively rinsed with filtered (0.2 µm) seawater and spun in a salad spinner to remove debris and epiphytes with this step being repeated multiple times. *Ulva* samples were weighed on a Scientech ZSA 120 digital microbalance (± 0.0001 g) to obtain initial wet weight in grams. All samples were kept in 100 mL 0.2 µm filtered seawater-filled containers after

spinning and weighing to prevent desiccation prior to use in experiments.

Small and large cohorts of *Mercenaria mercenaria* (~1 mm and ~5 mm, respectively) and *Argopecten irradians* (~5 mm and ~20 mm, respectively) used during experiments were spawned at the Stony Brook Marine Science Center of Stony Brook University hatchery (40.89° N, 72.44° W) using broodstock from Shinnecock Bay collected one-to-two months prior to spawning and exposed to environmental conditions (salinity, dissolved oxygen, pH) similar to their collection site. Small

and large cohorts of *Crassostrea virginica* (~2 mm and ~20 mm, respectively) used during experiments were produced by the Cornell Cooperative Extension shellfish hatchery, NY, USA (40.04° N, 72.39° W) using broodstock from the Peconic Estuary, NY, USA. Cohorts of small juvenile *Mytilus edulis* (~5 mm) used during experiments were collected from Shinnecock Bay, NY, USA during low tide (40.84° N, 72.50° W). Experiments using smaller bivalves (1 – 5 mm) were performed in 1 L polycarbonate vessels with 20 individuals per vessel, while experiments with larger bivalves (20 – 21 mm)

were performed in larger, 8 L polycarbonate vessels with five individuals per vessel.

Experiments began with the introduction of bivalves, *Ulva*, and nutrients into experimental vessels, with discrete measurements of pH and continuous measurements of dissolved oxygen and temperature made as described above throughout experiments. At the beginning of each experiment, 20 individuals from each bivalve cohort were set aside to obtain initial measurements of shell length (defined here as distance from umbo to furthest ventral margin), tissue weight,

and shell weight. Bivalve dimensions were determined via digital calipers and digital images with the two approaches producing nearly identical and not statistically different measurements. Captured images of bivalves were analyzed using ImageJ, with the scale of each image individually calibrated. Every three to four days, a complete water change was performed for all containers using water bubbled in 20-L carboys with gas mixtures for ambient and elevated $CO_2$ treatments as described above to ensure bivalves were exposed to their respective $CO_2$ concentrations. Once weekly, *Ulva* disks from

each container were removed, rinsed, spun in the salad spinner, weighed, and returned to the vessels. Additionally, every week, bivalves were collected on a 500 µm sieve, transferred to a petri dish, and measured for length with any mortality noted. Mortality rates were very low (always <10%) and did not differ among treatments. At the conclusion of experiments, final pH, temperature, and salinity measurements were made and final water samples for DIC analysis were collected and analyzed as described above. Additionally, 50 mL samples were removed from each container to assess final cell concentrations of phytoplankton provided for food (*I. galbana* and *C. muelleri*) which were preserved with Lugol's iodine (5%) solution and enumerated via microscopy (Tables 1 and S1).

At the conclusion of experiments, measurements of shell length for bivalves within the experimental containers as well as individuals set aside for initial measurements were made, and growth (expressed as $mm\,d^{-1}$) was determined from the changes in shell dimensions during the experiment. Tissue and shell weight were obtained by weighing bivalves after drying at 60°C for 72 hr, combusting them at 450°C for 4 hr, and weighing them again. Growth (expressed as $mg\,d^{-1}$) was determined by comparing the initial and final dry and combusted weights of individuals from each replicated vessel. Specifically, tissue weight was determined by subtracting the combusted weight from the dry weight, while shell weight was determined by subtracting the tissue weight from the dry weight. Two-way ANOVAs were performed using within SigmaPlot 11.0 to assess significant differences in growth rates based on shell length, tissue weight, shell weight, and survival during experiments, where the main treatment effects were $pCO_2$ (ambient or elevated), and the presence of *Ulva*. All data were log transformed prior to Two-way ANOVA to ensure that the assumptions of equal variance and normality were met. Normality was tested via the use of Shapiro-Wilk tests0. If significant differences were detected, a Tukey Honest Significant Difference (Tukey HSD) test using R 3.4.0 within RStudio 1.0.143 was performed to identify specific differences among treatments. Finally, linear regression models of shell length-, tissue weight-, and shell weight-based growth rates with $\Omega_{calcite}$ and $\Omega_{aragonite}$ were created using R-® software (version: 3.4.0; http://www.r-project.org).

## 3 Results

### 3.1 *Mercenaria mercenaria*

For the cohort of smaller juvenile *M. mercenaria* ($1.34 \pm 0.24$ mm), $\Omega_{calcite}$ and $\Omega_{aragonite}$ were significantly lower in treatments with elevated $CO_2$ (Two-way ANOVA; $p<0.001$ for both, Fig. 1; Tables S2-S3) and significantly higher in treatments containing *Ulva* (Two-way ANOVA; $p=0.002$ and $p=0.007$, respectively). Growth of the small *M. mercenaria* based upon shell length, shell weight, and tissue weight was highly sensitive to increases in $pCO_2$ as well as the presence of *Ulva*. When exposed to elevated $CO_2$ conditions, shell length-, shell weight-, and tissue weight-based growth rates were 49%, 66%, and 41% lower, respectively, when compared to their counterparts in ambient $CO_2$ treatments (Two-way ANOVA; $p<0.001$, $p<0.001$, and $p=0.038$, respectively; Fig. 1; Tables S4-S6). In contrast, shell length-, shell weight-, and tissue weight-based growth rates were significantly higher in the presence of *Ulva* (Two-way ANOVA; $p=0.006$, $p=0.011$, and $p=0.008$, respectively ; Fig. 1; Tables S4-S6) with growth based on shell length, tissue weight, and shell weight being 28%, 37%, and 47% higher, respectively, within elevated $CO_2$ treatments, and 10%, 25%, and 30%, respectively, within

ambient $CO_2$ treatments (Fig. 1). Multiple comparison tests revealed that *Ulva* often mitigated the negative effects of elevated $CO_2$ on hard clams. For example, length-based growth in elevated $CO_2$ treatments with *Ulva* was significantly higher than elevated $CO_2$ treatments without *Ulva* (Tukey HSD; $p=0.044$; Table S7). Furthermore, shell length-based growth rates showed strong, significant positive correlations with and $\Omega_{aragonite}$ and $\Omega_{calcite}$ across all treatments ($R^2=0.79$; $p<0.001$, and $R^2=0.79$; $p<0.001$, respectively; Table S10). There were also significant correlations between shell weight-based growth and $\Omega_{aragonite}$ ($R^2=0.53$; $p=0.001$; Table S10) and $\Omega_{calcite}$ ($R^2=0.53$; $p=0.002$; Table S11). For tissue weight-based growth, there were also significant correlations with $\Omega_{aragonite}$ and $\Omega_{calcite}$ ($R^2=0.30$; $p=0.05$ and $R^2=0.30$; $p=0.05$, respectively; Tables S10-S11).

For the larger-sized cohort of *M. mercenaria* ($5.00 \pm 0.41$ mm), $\Omega_{calcite}$ and $\Omega_{aragonite}$ were significantly higher in treatments containing *Ulva* (Two-way ANOVA; $p=0.002$ and $p<0.001$, respectively; Fig. 2; Tables S2-S3) and significantly lower in high $CO_2$ treatments (Two-way ANOVA; $p<0.001$ for both). Larger *M. mercenaria* responded to elevated $CO_2$ conditions and the presence of *Ulva* in a manner similar to that of the smaller clams. Under elevated $CO_2$ concentrations, shell length-, shell weight-, and tissue weight-based growth rates were significantly lower (by 45%, 30%, and 22%, respectively) relative than the ambient $CO_2$ treatments (Two-way ANOVA; $p=0.010$, $p=0.010$, and $p<0.001$, respectively; Fig. 2; Tables S4-S6). In the presence of *Ulva*, however, shell length-, shell weight-, and tissue weight-based growth rates were significantly higher by 10%, 21%, and 20%, respectively, in elevated $CO_2$ treatments, and by 21%, 18% , 162%, respectively, in ambient $CO_2$ treatments that did not receive *Ulva* (Two-way ANOVA; $p=0.003$, $p=0.006$, and $p=0.009$, respectively ; Fig. 2; Tables S4-S6). Across all treatments, shell length- and tissue weight-based growth rates were positively correlated with $\Omega_{aragonite}$ ($R^2=0.45$; $p=0.006$ and $R^2=0.44$; $p=0.013$, respectively; Table S10) and $\Omega_{calcite}$ ($R^2=0.45$; $p=0.006$ and $R^2=0.44$; $p=0.013$, respectively; Table S11). For shell weight-based growth, there were positive, nearly significant correlations with $\Omega_{aragonite}$ and $\Omega_{calcite}$ ($R^2=0.28$; $p=0.063$ and $R^2=0.28$; $p=0.063$, respectively; Tables S10-S11).

### 3.2 *Crassostrea virginica*

During the experiment with the cohort of small *C. virginica* ($2.45 \pm 0.41$ mm), $\Omega_{calcite}$ and $\Omega_{aragonite}$ were significantly higher in treatments containing *Ulva* (Two-way ANOVA; $p=0.025$ for both; Fig. 3; Tables S2-S3) and significantly lower in treatments receiving elevated $CO_2$ (Two-way ANOVA; $p<0.001$ for both). Growth rates of small *C. virginica* were sensitive to elevated $CO_2$ concentrations and the presence of *Ulva*. Length-, tissue-, and shell weight-based growth rates were 63%, 78%, and 145% lower, respectively, when exposed to elevated $CO_2$ concentrations compared to control treatments (Two-way ANOVA; $p=0.011$, $p=0.006$, and $p=0.012$, respectively; Fig. 3; Tables S4-S6). When in the presence of *Ulva*, shell length-based growth was significantly increased by 24% and 55% in elevated and ambient $CO_2$ treatments, respectively (Two-way ANOVA; $p=0.040$; Fig. 3; Table S4), but tissue and shell weight-based growth were not significantly different than the control (Two-way ANOVA; $p=0.319$ and $p=0.946$, respectively). Across all experimental vessels, there were significant positive correlations between shell length-, tissue weight-, and shell weight-based growth and $\Omega_{aragonite}$ ($R^2=0.26$;

$p=0.044$, $R^2=0.53$; $p=0.003$, and $R^2=0.39$; $p=0.013$, respectively; Table S10) and $\Omega_{calcite}$ ($R^2=0.26$; $p=0.045$, $R^2=0.53$; $p=0.003$, and $R^2=0.39$; $p=0.013$, respectively; Table S11).

For the larger juvenile *C. virginica* (24.92 ± 0.89 mm), $\Omega_{calcite}$ and $\Omega_{aragonite}$ were significantly higher in treatments containing *Ulva* (Two-way ANOVA; $p<0.001$ for both; Fig. 4; Tables S2-S3) and significantly lower in treatments receiving elevated $CO_2$ (Two-way ANOVA; $p<0.001$ for both). Growth responses for the larger *C. virginica* differed from the smaller-sized juveniles. Shell length-based growth was 167% significantly lower under elevated $CO_2$ concentrations relative to the control and significantly higher (by 23% and 450% in ambient and elevated $CO_2$ treatments, respectively) in the presence of *Ulva* relative to the control (Two-way ANOVA; $p=0.001$ and $p=0.006$, respectively ; Fig. 4; Table S4). While shell weight-based and tissue weight-based growth were not significantly altered by elevated $CO_2$ or the presence of *Ulva*, there was an antagonistic, interactive effect between both variables whereby the co-exposure to elevated $CO_2$ and *Ulva* yielded growth rates higher than would have been predicted by growth rates within the individual treatments (Two-way ANOVA; $p=0.024$; Fig. 4; Tables S5-S6). Consistent with this finding, shell length-based growth in elevated $CO_2$ treatments with *Ulva* was significantly higher than in elevated $CO_2$ treatments without *Ulva* (Tukey HSD; $p=0.032$; Table S7). There was a strong positive correlation between shell length-based growth and $\Omega_{aragonite}$ ($R^2=0.66$; $p=0.002$, respectively; Table S10) and $\Omega_{calcite}$ ($R^2=0.66$; $p=0.002$, respectively; Table S11) but not for tissue and shell weight-based growth.

### 3.3 *Argopecten irradians*

For the cohort of small *A. irradians* (4.73 ± 0.59 mm), $\Omega_{calcite}$ and $\Omega_{aragonite}$ were significantly higher in treatments containing *Ulva* (Two-way ANOVA; $p<0.001$ for both; Fig. 5; Tables S2-S3) and significantly lower in treatments with elevated $CO_2$ (Two-way ANOVA; $p<0.001$ for both). The growth of small juvenile *A. irradians* was altered by $pCO_2$ and, to a lesser extent, the presence of *Ulva*. Shell length-, tissue weight-, and shell weight-based growth rates were significantly reduced by exposure to elevated $CO_2$ concentrations (Two-way ANOVA; $p<0.001$, $p=0.023$, and $p=0.041$, respectively; Fig. 5; Tables S4-S6). Specifically, growth rates based on shell length, tissue weight, and shell weight were 26%, 40%, and 43% lower, respectively, when exposed to elevated $CO_2$ compared to ambient $CO_2$ treatments (Fig. 5). Shell length-based growth was significantly higher (by 10% and 29% in ambient and elevated $CO_2$ treatments, respectively) in the presence of *Ulva* relative to treatments that did not receive *Ulva* (Two-way ANOVA; $p=0.007$; Fig. 5; Table S4). In contrast, tissue and shell weight-based growth were not significantly affected by the presence of *Ulva* (Two-way ANOVA; $p=0.274$ and $p=0.637$, respectively; Fig. 5; Tables S5-S6). Shell length-based growth within elevated $CO_2$ treatments with *Ulva* was significantly higher than in the elevated $CO_2$ treatments without *Ulva* (Tukey HSD; $p=0.011$; Table S7). There were no significant differences in shell or tissue weight-based growth among any treatments (Tukey HSD; $p>0.05$ for all; Tables S8-S9). Comparisons within individual treatments showed that shell length-based growth within elevated $CO_2$ treatments without *Ulva* was significantly lower than the elevated $CO_2$ treatments with *Ulva* (Tukey HSD; $p=0.011$; Table S7). For all treatments, there were significant correlations between shell length-, tissue weight-, and shell weight-based growth of

smaller scallops and $\Omega_{aragonite}$ ($R^2$=0.56; $p$=0.001, $R^2$=0.36; $p$=0.018, and $R^2$=0.47; $p$=0.004, respectively; Table S10) and $\Omega_{calcite}$ ($R^2$=0.56; $p$=0.001, $R^2$=0.36; $p$=0.018, and $R^2$=0.47; $p$=0.004, respectively; Table S11).

For the larger cohorts of juvenile *A. irradians* (21.08 ± 1.06 mm), $\Omega_{calcite}$ and $\Omega_{aragonite}$ were significantly lower in treatments exposed to high $CO_2$ and significantly higher in treatments containing *Ulva* (Two-way ANOVA; $p<0.001$ for all;

Fig. 6; Tables S2-S3). The growth rates of larger *A. irradians* based on shell length and tissue weight were significantly reduced under elevated $CO_2$ concentrations by 32% and 105%, respectively (Two-way ANOVA; $p<0.001$ and $p$=0.019, respectively; Fig. 6; Tables S4 and S6) while shell weight-based growth was not (Two-way ANOVA; $p$=0.553; Table S5). Growth rates based on shell length and tissue weight were significantly increased in the presence of *Ulva* by 16% and 16%, respectively, in elevated $CO_2$ treatments, and by 60% and 16%, respectively, in ambient $CO_2$ treatments (Two-way ANOVA;

$p$=0.016 and $p$=0.032, respectively; Fig. 6; Tables S4 and S6) while shell weight-based growth was not (Two-way ANOVA; $p$=0.390; Table S5). There was a strong positive correlation between shell length-based growth of larger scallops and $\Omega_{aragonite}$ ($R^2$=0.74; $p$=0.001, respectively; Table S10) and $\Omega_{calcite}$ ($R^2$=0.74; $p$=0.001, respectively; Table S11) but not for tissue and shell weight-based growth.

### 3.4 *Mytilus edulis*

During the experiments with *M. edulis* (4.87 ± 0.92 mm), $\Omega_{calcite}$ and $\Omega_{aragonite}$ were significantly higher in treatments containing *Ulva* (Two-way ANOVA; $p$=0.017 and $p$=0.020, respectively; Fig. 7; Tables S2-S3) and significantly lower in treatments exposed to high $CO_2$ (Two-way ANOVA; $p<0.001$ for both). Growth rates of *M. edulis* based on shell length, tissue weight, and shell weight were all not significantly changed by exposure to elevated $CO_2$ concentrations (Two-way

ANOVA; $p$=0.149, $p$=0.210, and $p$=0.439, respectively; Fig. 7; Tables S4-S6). In contrast, shell length-, tissue weight-, and shell weight-based growth measurements were significantly higher in the presence of *Ulva* (Two-way ANOVA; $p$=0.045, $p$=0.047, and $p$=0.024, respectively; Fig. 7; Tables S4-S6). Specifically, in the presence of *Ulva*, growth based on shell length, tissue weight, and shell weight was 16%, 30%, and 45% higher, respectively, in elevated $CO_2$ treatments, and 28%, 19%, and 36%, respectively, in ambient $CO_2$ treatments relative to treatments that did not receive *Ulva* (Fig. 7). Mussel

growth rates were not correlated with $\Omega_{aragonite}$ or $\Omega_{calcite}$ (Tables S10-S11).

### 3.5 *Ulva* and microalgae

Across all experiments, the growth of *Ulva* was found to be significantly higher by 20% when exposed to elevated $CO_2$ concentrations (One-way ANOVA; $p$=0.043; Fig. S1; Table S12). Concentrations of *Isochrysis galbana* and

*Chaetoceros muelleri* were not significantly different between any treatment in any experiments (Two-way ANOVA; $p<0.05$ for all; Table S12). On average, final cell concentrations within treatments were ~90,000 cells $mL^{-1}$ (Tables 1 and S1).

## 4 Discussion

During this study, elevated $CO_2$ concentrations significantly reduced at least one or more growth measurements of cohorts of small- and large-sized juvenile *Mercenaria mercenaria*, *Crassostrea virginica*, and *Argopecten irradians*, but not *Mytilus edulis*. The presence of *Ulva* significantly increased the growth of all cohorts of all bivalve species. Comparisons of individual treatments indicated that under elevated $CO_2$ concentrations, the addition of *Ulva* often significantly increased growth rates of clams, scallops, and oysters by $23 - 30\%$. Both $\Omega_{aragonite}$ and $\Omega_{calcite}$ were significantly higher in the presence of *Ulva* in all experiments under both high and low $CO_2$ regimes, despite the rapid turnover of dissolved gas pools in experiments (>1000 time per day), and the growth rates of bivalves were significantly correlated with $\Omega_{aragonite}$ and $\Omega_{calcite}$ in treatment vessels for six of seven experiments. Collectively, these findings provide insight regarding the ability of macroalgae such as *Ulva* to mitigate the deleterious effects of ocean acidification on bivalves, and, potentially, other calcifying organisms.

The negative effects of ocean acidification on the growth and survival of bivalves and other calcifying organisms have been well-documented. Consistent with prior studies that have gauged the response of juvenile bivalves to elevated $CO_2$ (Gazeau et al., 2007; Green et al., 2009; Talmage and Gobler, 2011), the results of the current study show decreased tissue growth as well as calcification in the form of shell length- and weight-based growth under acidified conditions, a finding consistent with significantly lower $\Omega_{aragonite}$ and $\Omega_{calcite}$ in elevated $CO_2$ treatments. Early life-stage bivalve shells are composed partly or completely of aragonite, making them vulnerable to undersaturation of aragonite (Carriker, 1996; Stenzel, 1964; Talmage and Gobler, 2009). While the formation of calcium carbonate is thermodynamically favored when $\Omega$ exceeds 1.0, biotic aragonite is less crystalline than nonbiogenic aragonite (Weiss et al., 2002) and studies of early life stage Pacific oysters have suggested that a $\Omega_{aragonite}$ exceeding 1.6 may be required to yield successful growth and survival (Barton et al., 2012). Similarly, Talmage and Gobler (2010) found that increases in $\Omega_{aragonite}$ within the saturated range ($\Omega_{aragonite}$ increases from 2.9 to 3.3) significantly increased the growth of early life stage *M. mercenaria* and *A. irradians*, a finding suggesting that acidification since pre-industrial time can depress the performance of these species. In the current study, growth rates of bivalves exposed to *Ulva* under ambient $pCO_2$ frequently exceeded those of individuals grown under the same $CO_2$ delivery rate without *Ulva* as $\Omega_{aragonite}$ was significantly increased, on average from 1.91 to 2.16 (Table 1), with both levels being saturated but also being below the threshold that yielded maximal growth rates in early life stage bivalves for Talmage and Gobler (2010). Furthermore, even minor, yet sustained, increases or decreases in $\Omega_{aragonite}$ (<0.1 units) can result in significant changes in the growth of larval and juvenile bivalves (Barton et al., 2012; Talmage and Gobler, 2011), which was similarly observed in many of the experiments in the present study. Hence, the potential benefits of macroalgae to calcifying bivalves may be realized in both acidified and 'normal' conditions.

Acidification can have cascading negative physiological consequences for bivalves. In larval bivalves, high $CO_2$ depresses calcification, lipid content, RNA:DNA ratios, metamorphosis, and growth rates (Gobler and Talmage, 2013). The reduction in tissue weight-based growth under elevated $CO_2$ concentrations found during the present study is consistent with Beniash et al. (2010), who found significant declines in soft body mass of juvenile *C. virginica* maintained in hypercapnia

(pH 7.5). Additionally, the same study and others (Gazeau et al., 2007; Matoo et al., 2013) have reported increased metabolic rates in bivalves exposed to elevated $CO_2$ levels. As suggested by Waldbusser et al. (2015b), decreasing $\Omega_{aragonite}$ increases the amount of energy spent by bivalves on shell formation which diverts energy away from maintaining homeostasis and other metabolic processes including those that contribute toward growth (Beniash et al., 2010; Waldbusser et al., 2015b).

Macroalgae can control carbonate chemistry in shallow ecosystems and, in turn, can affect the performance of carbonaceous organisms. A study by Anthony et al. (2013) found that within mixed assemblages of turf and fleshy macroalgae, the saturation state of aragonite increased during the daytime. Krause-Jensen et al. (2015) reported that macroalgae may provide a refuge for calcifying organisms. Specifically, within a subarctic fjord, macroalgae drove strong diel variability in pH and $\Omega_{aragonite}$, with *M. edulis* being found to grow in close association with macroalgae, even in tidal

pools that became supersaturated and undersaturated between day and night cycles, respectively (Krause-Jensen et al., 2015). Additionally, Wahl et al. (2017) demonstrated that daytime increases in pH associated with the macroalgae *Fucus vesciculosus* provided a refuge against acidified conditions for *M. edulis*, with calcification rates of *M. edulis* increasing with increases in pH wrought by the algae. In the current study, *Ulva* yielded significantly increased $\Omega_{aragonite}$, $\Omega_{calcite}$, and bivalve growth in all seven experiments performed. Dissolved oxygen levels were also high (> 8 mg L$^{-1}$) in all treatments and the

growth rates of bivalves were often significantly higher in high $CO_2$ treatments with *Ulva* compared to those without. Furthermore, in ambient and elevated $CO_2$ treatments, the presence of *Ulva* significantly increased pH beyond levels observed in treatments without *Ulva*, often in as little as 24 hr, with those increases sustained over duration of the experiments (Figs. S2-S3). Hence, it seems likely that the macroalgae buffered carbonate chemistry to the benefit of bivalves. While it is possible that photosynthetic activity by microalgae (*I. galbana* and *C. muelleri*) may have contributed to

shifts in carbonate chemistry, there were no significant differences in microalgae cell concentrations in any treatment across all experiments (Two-way ANOVA; $p>0.05$; Table S13) suggesting that microalgal contributions to changes in carbonate chemistry were minimal relative to the photosynthetic activity of *Ulva*.

        Beyond photosynthesis, macroalgae may also alter carbonate chemistry via the uptake of nitrogenous nutrients. Specifically, the uptake of nitrate or ammonium by marine autotrophs results in an equimolar increase or decrease in total

alkalinity, respectively (Brewer and Goldman, 1976; Goldman and Brewer, 1980; Talmage and Gobler, 2012), which occurs due to the production of OH$^-$ and H$^+$ to balance the uptake of nitrate and ammonium, respectively (Brewer and Goldman, 1976; Goldman and Brewer, 1980; Redfield et al., 1963). Given that 50µM nitrate was added to all experimental vessels with *Ulva* to promote its growth during each experimental water change, it is possible that the assimilation of this nitrate by *Ulva* contributed to the average 10 – 20 µM increase in total alkalinity observed within treatments with *Ulva* (Two-way ANOVA;

$p<0.05$; Table 1; Tables S14-S15). Higher alkalinity seawater requires higher concentrations of $CO_2$ to reduce pH, thus resulting in smaller changes in $\Omega_{aragonite}$ and $\Omega_{calcite}$. Given the rapid turnover of dissolved gasses in experimental vessels, it is possible the nitrogen assimilation effects on alkalinity outweighed the effects of photosynthetic consumption of DIC.

        Prior studies have found that *Ulva* can experience enhanced growth (Björk et al., 1993; Olischläger et al., 2013; Young and Gobler, 2016) and outcompete other autotrophs (Young and Gobler, 2017) under elevated $CO_2$ concentrations.

Hence, the dominance of *Ulva* and similar macroalgae in estuaries that experience seasonal acidification (Wallace and Gobler, 2015) could ultimately benefit bivalves and other calcifying organisms. In the present experiments, *Ulva* growth was, on average, ~20% higher under elevated $CO_2$ conditions. Furthermore, the presence of the macroalgae frequently transformed $\Omega_{aragonite}$ of elevated $CO_2$ treatments from undersaturated to nearly saturated (Tables 1 and S1) and often yielded

growth rates of bivalves significantly greater than the elevated $CO_2$ treatments without *Ulva*. Had the dissolved gas pools within experimental vessels not been turned over rapidly via aeration, it is possible the effects of *Ulva* on the carbonate chemistry would have been even greater.

The benefits of *Ulva* and detriments of high $CO_2$ to the four bivalves studied differed by species. While every cohort displayed significantly enhanced growth in the presence of *Ulva*, scallops were the only species to experience

significantly higher growth in the elevated $CO_2$ treatment with *Ulva* compared to the elevated $CO_2$ treatment without *Ulva* for both the small and large juvenile cohorts. In contrast, for clams and oysters, only one of the two cohorts displayed this specific response. Early life stages of bay scallops have been consistently shown to be more vulnerable to acidification than the other bivalve species studied here (Stevens and Gobler, in revision; Talmage and Gobler, 2009, 2011). This may be due, in part, to its rapid growth and metabolism compared to other bivalves (Kennedy et al., 1996; Kraeuter and Castagna, 2001;

Shumway and Parsons, 2006), traits that may also make it more likely to benefit from the improved carbonate chemistry wrought by the presence of *Ulva*. The resistance of *M. edulis* to elevated $CO_2$ concentrations contrasted with prior studies of European strains of this bivalve (Berge et al., 2006; Gazeau et al., 2007) but is consistent with prior cohorts of this species isolated from Shinnecock Bay, NY, USA (Stevens and Gobler, in revision). However, Thomsen et al. (2013) found that specific growth and calcification rates of juvenile *M. edulis* under acidified conditions were dependent on food availability.

Given that food was supplied *ad libitum* in the present study, it is possible that the negative effects of elevated $CO_2$ concentrations on *M. edulis* may have been mitigated by adequate food availability as well as improved carbonate chemistry facilitated by *Ulva*.

While growth rates of bivalves were significantly correlated with $\Omega_{aragonite}$ and $\Omega_{calcite}$ during this study, the absolute change in bivalve growth was greater than what might be expected from the observed *Ulva*-induced changes in $\Omega_{aragonite}$ and

$\Omega_{calcite}$ suggesting others factors may have contributed to the increased growth of bivalves in treatments with *Ulva*. While *Ulva* increased dissolved oxygen levels in vessels, dissolved oxygen concentrations were high in all treatments ($> 8$ mg $L^{-1}$). Adequate food availability makes bivalves more resilient against ocean acidification (Thomsen et al., 2013) but microalgal food levels were always in excess and did not different across treatments in our experiments (Table S13). While we cannot discount the possibility that *Ulva* supplied an unknown nutritional factor that benefited the growth of the bivalves, this does

not seem likely given the known particle capture range of juvenile bivalves (~3 – ~100 µm; Ward and Shumway, 2004). A more important factor to consider is the time of day during which discrete pH measurements were made during this study. Limited, continuous measurements of pH in vessels with *Ulva* made during this study revealed a strong diurnal pattern for pH values driven by photosynthesis, with minimal levels in the early morning and peak values in the evening (Fig. S4). The discrete, daily measurements of pH made in vessels that were used to determine the carbonate chemistry during this study

were made during the late morning when pH values were low relative to the complete daily cycle (Fig. S4). Had our carbonate chemistry determinations been based on maximal pH values present during the late evening, we estimate that $\Omega_{aragonite}$ and $\Omega_{calcite}$ values would have been ~0.4 and ~0.6 units higher, respectively, than reported (Tables S1 and Fig. S4). Future studies that make continuous measurements of pH and DIC or measurements that represent the mean conditions present within vessels will be able to better represent the precise carbonate chemistry conditions that *Ulva* creates that, in turn, promotes the growth of bivalves. Regardless, even with $\Omega_{aragonite}$ and $\Omega_{calcite}$ values that were likely underestimated, this study demonstrated the ability of *Ulva* to significantly increase $\Omega_{aragonite}$, $\Omega_{calcite}$, and bivalve growth rates as well as the significant correlation between $\Omega$ and bivalve growth rates.

Beyond the modification of carbonate chemistry, there are additional ecosystem benefits that may be provided by macroalgae. Macroalgal beds can serve as a nursery habitat for juvenile *Callinectes sapidus* (Wilson et al., 1990), as well as other decapods (Heck et al., 2003; Sogard and Able, 1991). Macroalgae can also serve as a refuge from predation for some juvenile and adult bivalves (Carroll et al., 2010). An additional potential benefit provided to bivalves by *Ulva* and other macroalgae is their ability to inhibit the growth of phytoplankton species that cause harmful algal blooms (HABs; Tang and Gobler, 2011; Tang et al., 2015) that can directly harm the bivalve species used in the present study (Gobler and Sunda, 2012; Leverone et al., 2006; Stoecker et al., 2008; Tang and Gobler, 2009). Furthermore, given its ability to rapidly assimilate and store nitrate and ammonium (Pedersen and Borum, 1997), *Ulva* can serve as a biofilter within eutrophic ecosystems (Hernández et al., 2002; Neori et al., 2003). Given that harmful algal blooms flourish in eutrophic zones (Anderson et al., 2008; Anderson et al., 2002), the mitigation of high nutrient conditions by *Ulva* may reduce the intensity of HABs, which may indirectly benefit bivalve species that are negatively impacted by the occurrence of such events. Finally, there is great precedent for the deployment of macroalgae as a principal component of integrated multi-trophic aquaculture systems whereby seaweeds are co-cultivated with aquacultured shellfish with the macroalgae often being harvested for profit (Neori, 2008; Nobre et al., 2010; Troell et al., 2009). Such an approach may be increasingly important for the protection of aquacultured bivalves in an increasing acidified ocean in the future.

Despite the reported positive interactions between *Ulva* and the various species of bivalves in prior studies (Carroll et al., 2010; Heck et al., 2003; Sogard and Able, 1991; Wilson et al., 1990; this study), macroalgae can negatively impact bivalves and other calcifying organisms. Secondary metabolites released by *Ulva* can elevate mortality rates in the larval stages of bivalves (Diederich, 2005; Nelson et al., 2003), barnacles (Brock et al., 2007; Magre, 1974), crabs (Johnson and Welsh, 1985), and molluscs (Wang et al., 2011). *Ulva* can form "green tides" (Smetacek and Zingone, 2013) that, upon their collapse, can create hypoxic regions (Valiela et al., 1992) that can negatively affect benthic fauna (Viaroli et al., 2001). Furthermore, extensive coverage of bivalves by *Ulva* and the subsequent decomposition of the algae can also result in the accumulation of $H_2S$, which, when coupled with low dissolved oxygen, can depress the growth and survival of bivalves (Tyler, 2007). However, as pointed out by Wilson et al. (1990), the accumulation of secondary metabolites and decreased dissolved oxygen associated with the overgrowth of *Ulva* is often mitigated in high-flow areas, alleviating potential harm to

the nearby organisms. Furthermore, it is likely that other macroalgae that are not known to negatively impact marine life provide similar buffering of carbonate chemistry (Anthony et al., 2013; Krause-Jensen et al., 2015; Wahl et al., 2017).

Numerous species of seagrass experience enhanced growth under elevated $CO_2$ concentrations (Beer and Koch, 1996; Palacios and Zimmerman, 2007; Zimmerman et al., 1997) and can buffer ocean acidification thus benefiting calcifying

organisms (Garrard et al., 2014; Hendriks et al., 2014). However, this ecosystem service may be disrupted by eutrophication (Valiela et al., 1997) and acidification (Young et al., in press) of coastal ecosystems which could favor the growth of macroalgae over seagrass (Young et al., in press). As seagrasses decline worldwide (Orth et al., 2006; Short et al., 2011), the ecosystem services provided by seagrasses, such as being nursery habitats or buffering against ocean acidification, may, in some cases, be provided by macroalgae, potentially benefiting calcifying organisms such as bivalves that had formerly

depended on seagrass as a refuge habitat.

In conclusion, during this study photosynthetic activity and/or nitrate assimilation by *Ulva* increased $\Omega_{aragonite}$ and $\Omega_{calcite}$ and yielded enhanced growth of bivalves by mitigating the deleterious effects of elevated $pCO_2$. This benefit was not exclusive to acidified conditions, as evidenced by increased bivalve growth in the presence of *Ulva* within ambient $CO_2$ treatments. While macroalgae can have adverse effects on some larval-staged bivalves, the chemical resilience provided by

the macroalgae, *Ulva*, along with other potential ecosystem benefits such as providing nursery habitat (Wilson et al., 1990), predation refuge (Carroll et al., 2010), and inhibiting the growth of harmful microalgae (Tang and Gobler, 2011; Tang et al., 2015) may, in some case, outweigh the negative effects. Given that macroalgae tend to outcompete seagrass under high $CO_2$ conditions (Young et al., in press), the ability of macroalgae to provide ecosystem services similar to those of seagrass, particularly buffering carbonate chemistry, may be increasingly important for calcifying organisms in modern-day eutrophic,

acidified estuaries, as well as within future, ocean acidification scenarios. Finally, the purposeful deployment of seaweeds in an aquaculture setting would seem to be a beneficial strategy for protecting bivalves against current and future acidification.

## 5 Author contributions

Conceived and designed the experiments: C.J.G., C.S.Y. Performed the experiments: C.S.Y. Analyzed the data:

C.S.Y., C.J.G. Contributed reagents/materials/analysis tools: C.J.G. Wrote the manuscript: C.S.Y., C.J.G.

## 6 Acknowledgements

We are grateful for the supply of *Crassostrea virginica* provided by the Cornell Cooperative Extension of Huntington, NY and Southold, NY. We thank Stephen Heck for the collection of *Mytilus edulis*. We are appreciative of the

logistical support provided by the Stony Brook Southampton Marine Science Center staff throughout this study. This work was supported by New York Sea Grant R-FMB-38 and grants from the Chicago Community Foundation, the Laurie Landaeu Foundation, and the Pritchard Foundation.

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

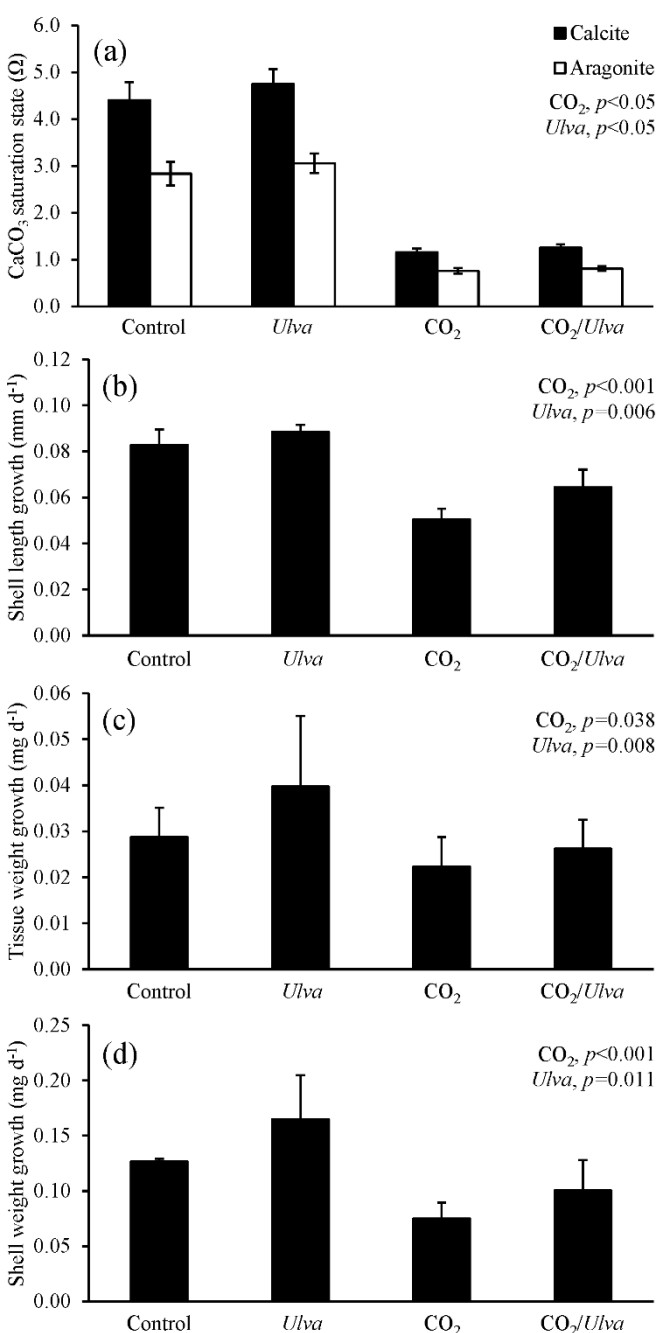

**Figure 1.** Experiment with small juvenile *Mercenaria mercenaria* exposed to ambient and elevated concentrations of $CO_2$ with and without the presence of *Ulva*.; (a) $\Omega_{calcite}$ and $\Omega_{aragonite}$; Growth was based on (b) shell length; (c) tissue weight; and (d) shell weight. Columns represent means ± standard deviation. Significant main treatment effects ($CO_2$ and *Ulva*) appear on the top right of each figure.

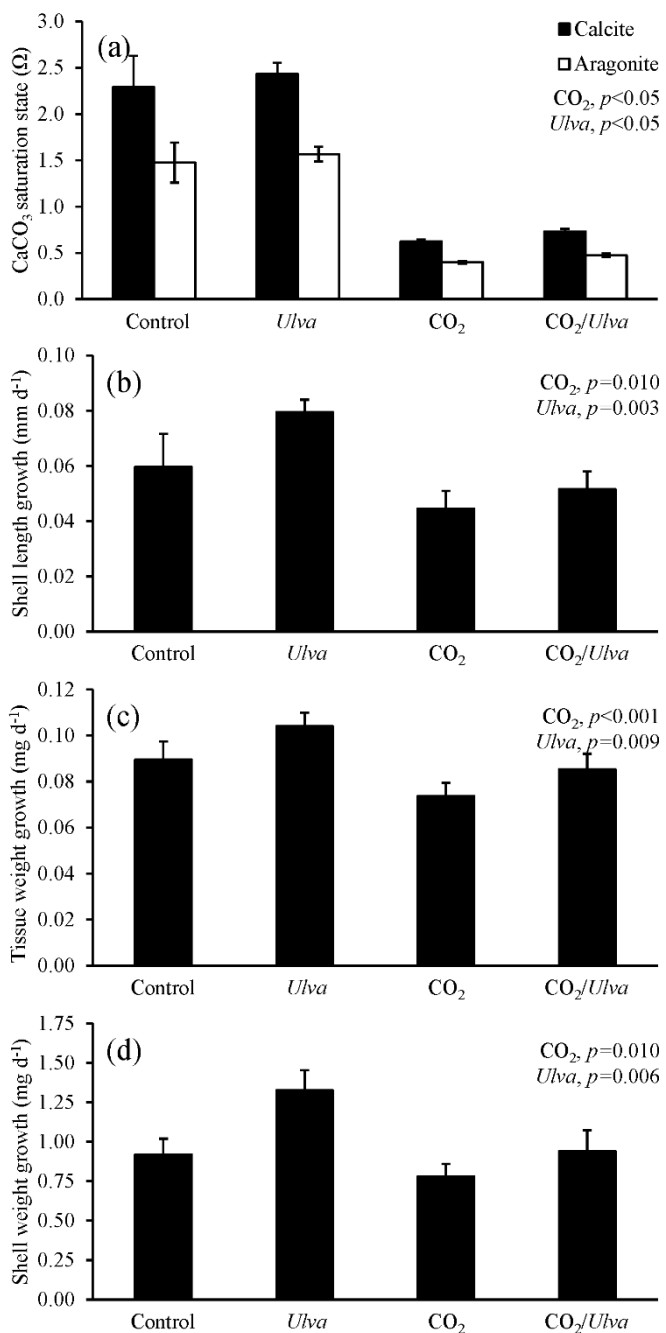

**Figure 2.** Experiment with large juvenile *Mercenaria mercenaria* exposed to ambient and elevated concentrations of $CO_2$ with and without the presence of *Ulva*. (a) $\Omega_{calcite}$ and $\Omega_{aragonite}$; Growth was based on (b) shell length; (c) tissue weight; and (d) shell weight. Columns represent means ± standard deviation. Significant main treatment effects ($CO_2$ and *Ulva*) appear on the top right of each figure.

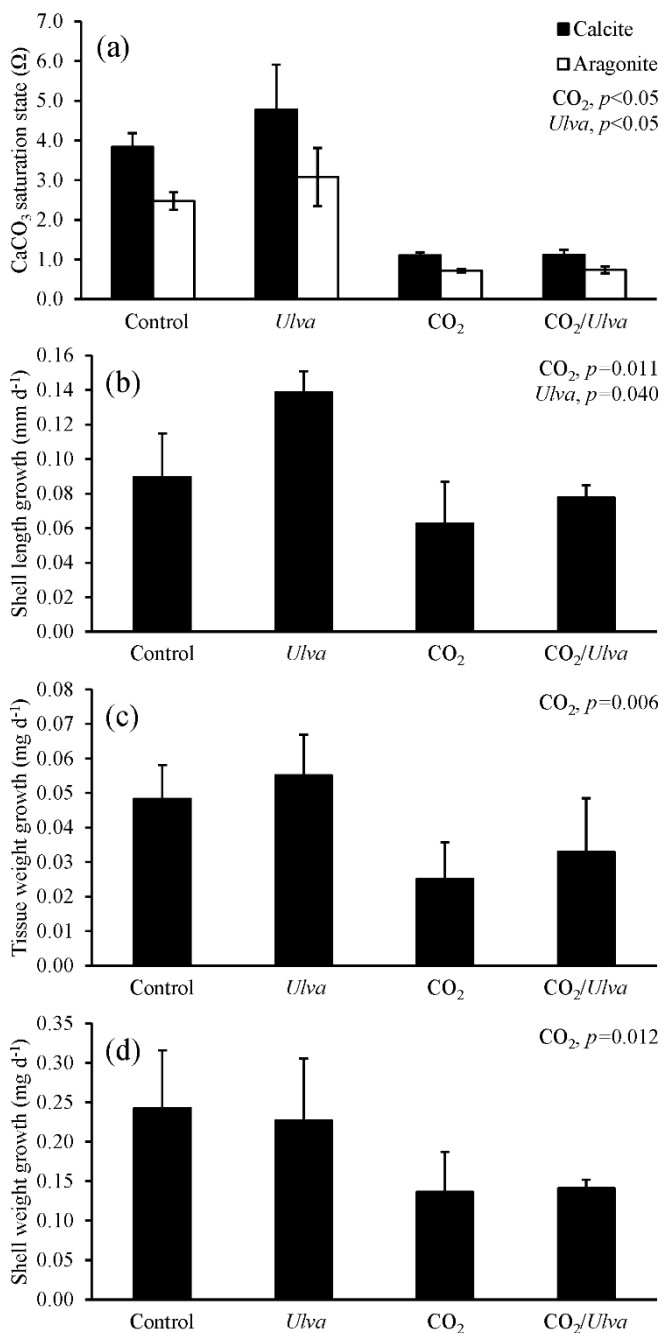

**Figure 3.** Experiment with small juvenile *Crassostrea virginica* exposed to ambient and elevated concentrations of $CO_2$ with and without the presence of *Ulva*. (a) $\Omega_{calcite}$ and $\Omega_{aragonite}$; Growth was based on (b) shell length; (c) tissue weight; and (d) shell weight. Columns represent means ± standard deviation. Significant main treatment effects ($CO_2$ and *Ulva*) appear on the top right of each figure.

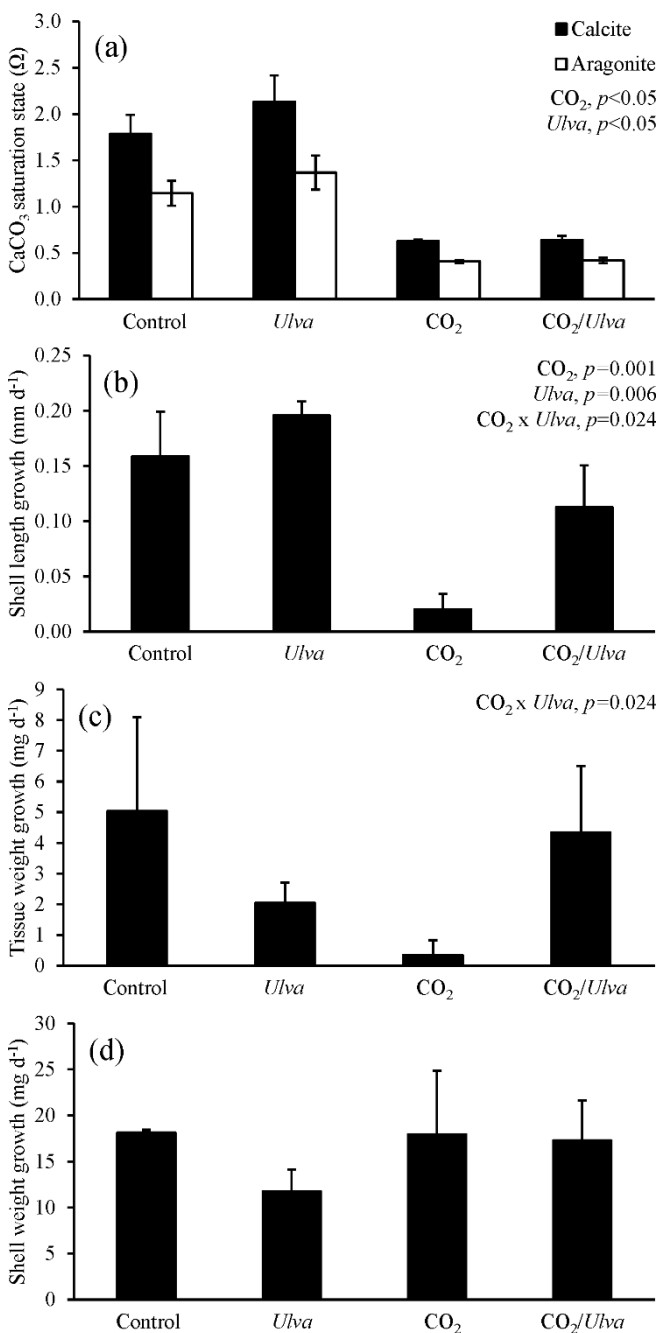

**Figure 4.** Experiment with large juvenile *Crassostrea virginica* exposed to ambient and elevated concentrations of $CO_2$ with and without the presence of *Ulva*. (a) $\Omega_{calcite}$ and $\Omega_{aragonite}$; Growth was based on (b) shell length; (c) tissue weight; and (d) shell weight. Columns represent means ± standard deviation. Significant main treatment effects ($CO_2$ and *Ulva*) appear on the top right of each figure.

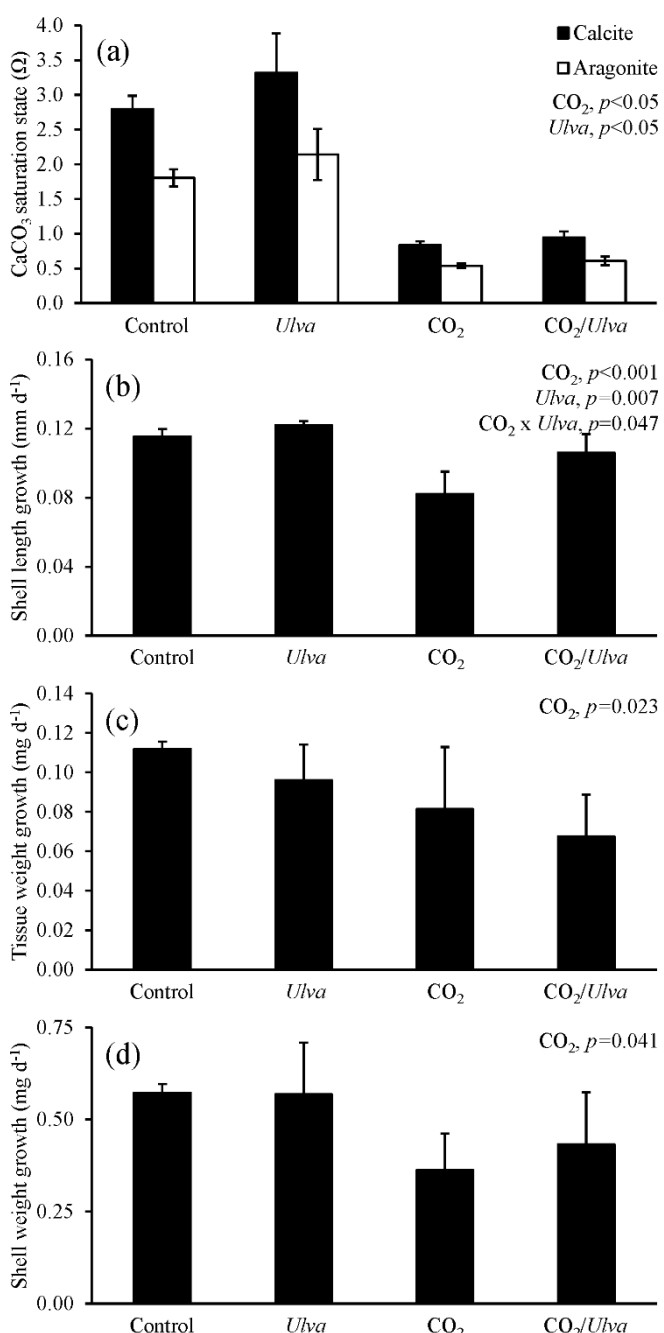

**Figure 5.** Experiment with small juvenile *Argopecten irradians* exposed to ambient and elevated concentrations of $CO_2$ with and without the presence of *Ulva*. (a) $\Omega_{calcite}$ and $\Omega_{aragonite}$; Growth was based on (b) shell length; (c) tissue weight; and (d) shell weight. Columns represent means ± standard deviation. Significant main treatment effects ($CO_2$ and *Ulva*) appear on the top right of each figure.

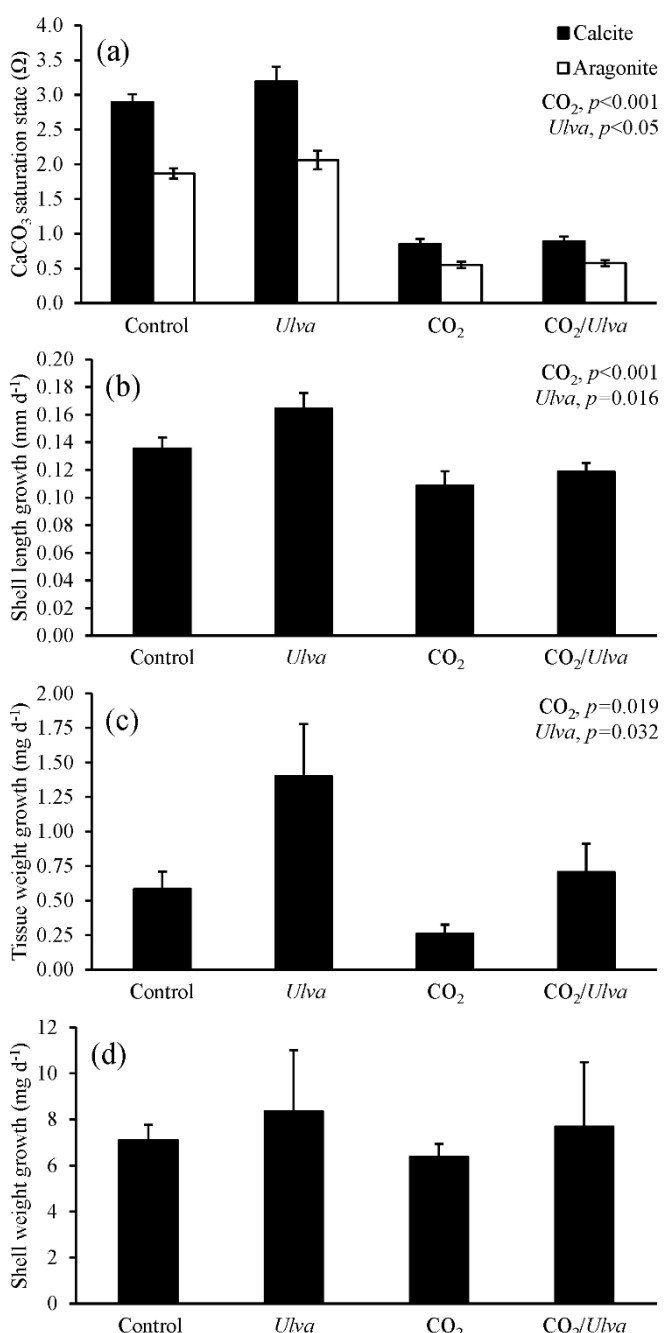

**Figure 6.** Experiment with large *Argopecten irradians* exposed to ambient and elevated concentrations of $CO_2$ with and without the presence of *Ulva*. (a) $\Omega_{calcite}$ and $\Omega_{aragonite}$; Growth was based on (b) shell length; (c) tissue weight; and (d) shell weight. Columns represent means ± standard deviation. Significant main treatment effects ($CO_2$ and *Ulva*) appear on the top right of each figure.

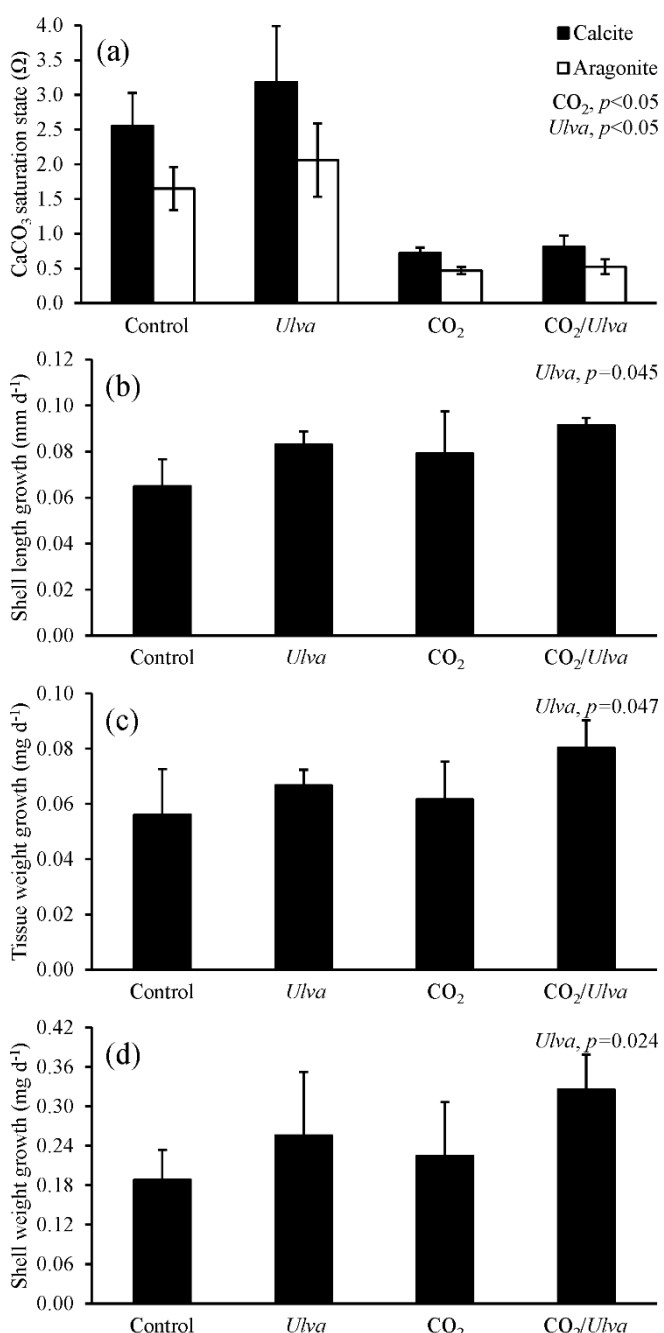

**Figure 7.** Experiment with *Mytilus edulis* exposed to ambient and elevated concentrations of $CO_2$ with and without the presence of *Ulva*. (a) $\Omega_{calcite}$ and $\Omega_{aragonite}$; Growth was based on (b) shell length; (c) tissue weight; and (d) shell weight. Columns represent means ± standard deviation. Significant main treatment effects ($CO_2$ and *Ulva*) appear on the top right of each figure.

**Table 1.** Values of mean pH (total scale), temperature (°C), dissolved oxygen (mg $L^{-1}$), and salinity (g $kg^{-1}$), and final $pCO_2$ (µatm), total alkalinity (µmol $kgSW^{-1}$), total DIC (µmol $kgSW^{-1}$), $HCO_3^-$ (µmol $kgSW^{-1}$), $CO_3^{2-}$ (µmol $kgSW^{-1}$), $OH^-$ (µmol $kgSW^{-1}$), $\Omega_{calcite}$, $\Omega_{aragonite}$, and final microalgal cell counts of *Isochrysis galbana* and *Chaetoceros muelleri* (cells $mL^{-1}$) for June through November experiments (n=4 for all treatments). Values represent means ± standard error. Asterisks indicate parameters that were directly measured, and not calculated. Data from individual experiments appear within Tables S1.

| Parameter | Control | *Ulva* | $CO_2$ | $CO_2$/*Ulva* |
|---|---|---|---|---|
| pH* | 7.98±0.01 | 8.03±0.01 | 7.37±0.01 | 7.39±0.01 |
| Temperature* | 21.3±0.1 | 21.2±0.1 | 21.3±0.1 | 21.3±0.1 |
| Dissolved oxygen* | 9.06±0.01 | 9.00±0.01 | 9.17±0.01 | 9.10±0.01 |
| Salinity* | 30.0±0.1 | 30.1±0.1 | 30.0±0.1 | 30.0±0.1 |
| $pCO_2$ | 373±8 | 335±9 | 1763±27 | 1721±27 |
| Total alkalinity | 1740±26 | 1759±26 | 1792±25 | 1803±21 |
| Total DIC* | 1561±19 | 1557±21 | 1782±22 | 1797±19 |
| $HCO_3^-$ | 1428±16 | 1413±18 | 1690±21 | 1706±19 |
| $CO_3^{2-}$ | 119±4 | 134±5 | 35±1 | 37±1 |
| $OH^-$ | 3.84±0.12 | 4.51±0.18 | 0.95±0.02 | 1.01±0.02 |
| $\Omega_{calcite}$ | 2.97±0.11 | 3.36±0.13 | 0.86±0.03 | 0.90±0.03 |
| $\Omega_{aragonite}$ | 1.91±0.07 | 2.16±0.09 | 0.56±0.02 | 0.59±0.02 |
| Microalgae cells* | 97273±5230 | 97727±4696 | 90455±4388 | 95000±5294 |