# Peer review of "The ability of macroalgae to mitigate the negative effects of ocean acidification on four species of North Atlantic bivalve"

_Biogeosciences, 2018_

## Referee Comment (RC1) · Anonymous Referee #1 · 15 May 2018

Two size cohorts of hard clams, oysters, scallops, and mussels, were grown with and without macroalga Ulva in two CO2 treatments. The results show higher growth rates of bivalves in presence of Ulva, with a small benefit in the high CO2 treatment. Increased bivalve growth rates in the presence of Ulva was attributed to the increase in saturation state caused by Ulva.

The study is an interesting approach to study the potential benefit of Ulva on growth of multiple bivalve species, in the context of aquaculture management with ocean acidification. The strength of this study is that the experiment was conducted on multiple

species, two size classes, and there are multiple growth metrics with consistent results. The weakness of this study is the seawater chemistry and the conclusions drawn from the data. The results are intriguing and merit further exploration of why bivalves exhibited enhanced growth in the presence of Ulva. As not all factors were controlled in this experiment (e.g. unknown effect of algae and mussels on seawater chemistry, independently and by treatment), this study provides results to further develop specific hypotheses as to why these trends were observed. In its current form, I am not convinced by the conclusion that Ulva alters seawater chemistry which in turn causes increased bivalve growth under high CO2.

Main concerns:

1. The authors attribute what is a substantial biological response by bivalves in the presence of Ulva and high CO2 to a very MINOR increase in saturation state over time (only 0.04!). A lot of emphasis is placed on statistical comparisons of saturation state across treatments, probably because the change is so small but offers an attractive explanation. However, a statistically significant difference in a carbonate chemistry parameter across treatments does not mean that it is biologically relevant. The authors do not discuss if the magnitude of change in growth is realistic for a 0.04 change in saturation state (perhaps some summary plot showing growth metrics of each species by treatment, with aragonite saturation state of each treatment on the x-axis, would provide insight). However, Comment #2 explains why the sampling design is insufficient to characterize seawater chemistry in this experiment in the first place.

2. The seawater chemistry sampling design and measurements are not sufficient to describe how organisms contributed to seawater chemistry or what they actually experienced.

a. Water was only sampled at the start and end of the experiment, despite multiple water changes during the closed-system experiment. If the changes in saturation state come from cumulative effect of nitrate assimilation by Ulva, this is in fact a change that

occurred since the last water change (every 3 days). It means that the bivalves mostly experienced the same saturation state across high CO2 treatments, regardless of the 0.04 change that would have occurred over 3 days.

b. Seawater chemistry was highly variable. According to the authors, Ulva changes carbonate chemistry via CO2 uptake (decreasing DIC; P9, L11-22) and/or nutrient uptake (increasing TA, estimated at 10-20 umol/kg; P9,L29). During the experiment, the effect of CO2 uptake via primary production by Ulva is presumably removed with continuous bubbling with treatment concentrations of air/CO2 gas mix (P9). However, pCO2 is quite variable across treatments and experiments, indicating that the method used for bubbling did not actually bring the system (treatments + biology) into equilibrium. For example, within one experiment, the standard error in pCO2 reported in Table S1 is up to 200 uatm (based on N=2, start and end samples?). TA also varied substantially, even across treatments without Ulva, and TA did not always increase in the presence of Ulva (Table S1, this is masked by Table 1 which somewhat deceptively summarizes treatments across all experiments). For example, TA was 230 umol/kg less in the CO2 treatment compared to control in the experiment for Mercenaria mercenaria, even without Ulva. The authors do not describe why all their measurements are so variable and inconsistent in what they define as a well-controlled system. It is unclear if SE refers to a start and end sampling, which again is not a relevant design if the authors think that biological processes contribute to changes in seawater chemistry.

c. Chemistry was calculated using pH that was measured by a Durafet but no information on calibration and quality control was provided. It is unclear how and where the daily pH measurements are used.

d. For all of the above reasons, I am not convinced that photosynthesis or nitrate assimilation by Ulva increased saturation state which then enhanced growth of bivalves (as claimed on P11, L29-30). Unless the authors can clarify these points, alternative hypotheses should be discussed. For example, could proliferation of algal cells in high CO2 have provided more food to the bivalves and therefore contributed to their growth?

After all, nutrients were added and this would benefit Isochrysis spp. (spelling error on P3,L23) and Chaetoceros spp.

3. The extensive discussion (e.g. last four paragraphs) on macroalgae/seagrass benefits to bivalves detracts from the discussion of the results of this study, and makes the authors appear biased towards the hypothesis that macroalgae will mitigate ocean acidification (e.g., their interpretation of Unsworth et al 2012 on P11,L17, comments below). The ability for seagrass and macroalgae to chemically buffer ocean acidification (e.g., P12, L1-2) is not a fact, and needs to be considered in the context of the greater coastal environment that the habitat is in (e.g., freshwater inputs, upwelling, water residence time, etc., e.g. see Cyronak at al 2018 "Short-term spatial and temporal carbonate chemistry variability in two contrasting seagrass meadows: implications for pH buffering capacities"). The authors do not discuss the fact that their experiment was conducted in a closed system. It is unrealistic to conclude that a minute impact on alkalinity by Ulva (if verified, see comment 1 & 2) would mitigate ocean acidification in an open system. For these reasons, extrapolating these results to field applications should not take up more than a paragraph, and the authors should only do so if all of the issues with seawater chemistry can be sufficiently resolved.

Minor comments:

Title: based on the issues with seawater chemistry, this title may need to be revised

Abstract: remove p-values

Introduction:

- Half of this study has to do with large vs. small bivalves but the significance of this is not mentioned in the introduction. Please add the motivation for this in the Introduction.

- P2, L18: specify that pH and saturation state in seagrass meadows provide *temporal* refuge from acidification (as pH also declines below background seawater pH at night or in winter seasons).

Methods:

- Were nutrients added to vessels without Ulva as well? If not, the presence of Ulva is confounded with presence of nutrients which could influence the growth of Isochrysis and Chaetoceros and therefore the food supply by treatment.

- P3, L 24: how can 'ad libitum' food supply be exactly 4 x 104 cells mL-1 d-1?

- Report on assumptions of ANOVA (i.e., do residuals exhibit a normal distribution? was this tested?)

- P4, L34: add # of circles of algae added to each vessel. Was this scaled by container volume for small (1 L) and large bivalves (8 L)? If Ulva changes seawater chemistry in a consistent way, this data can be used to explore that (e.g., weight to volume and magnitude change in seawater chemistry).

Results:

- Tables in supplement: check consistency of * with p<0.05.

- Please report the actual p-values in the text since the tables are in supplemental files.

- I don't understand how ANOVA results are used to make statements like "When in the presence of Ulva, shell length-based growth was significantly increased by 42% (Two-way ANOVA; p<0.05)" when it is unlikely that the % change is the same in high CO2 and low CO2 treatments. If the authors are reporting the effect of Ulva only at high CO2, then the statistics should come from the Tukey post-hoc comparison. Authors should also report on the interaction of the two-way ANOVA (significant or not).

- I was expecting the Ulva results in the Results section. It's not critical, but a small point of confusion.

- P5, L35: report tests of ANOVA assumptions, report p-values that are corrected for multiple comparisons.

Discussion:

- P11,L16-19: this statement is incorrect. Unsworth et al 2012 is a theoretical modeling study. Model results were then applied to coral calcification rates that came from laboratory-based experiments. The authors themselves state that the results from the modeling need to be field tested.

- Discussion should include information about the magnitude of the beneficial effect of Ulva under high CO2.

Table 1: indicate which parameters were measured, and sample size (N).

Figures: define error bars and indicate when there are significant differences among groups

---

## Referee Comment (RC2) · Anonymous Referee #2 · 21 May 2018

"The ability of macroalgae to mitigate the negative effects of ocean acidification on four species of North Atlantic bivalve"

This paper evaluates the effect of the presence of the macroalga Ulva rigida on the growth of four North Atlantic bivalve species, Mercenaria mercenaria, Crassostrea virginica, Argopecten irradians and Mytilus edulis. The authors have used small and larger sizes of three out of four species, specifically the three obtained from hatcheries. The pCO2 levels the bivalves are exposed to are high, but conceivable for estuarine systems. The authors claim that "saturation states for calcium carbonate ($\Omega$) were

significantly higher in the presence of Ulva under both ambient and elevated CO2 delivery rates (p<0.05)", and that "alkalinity was increased by the presence of Ulva". This might be statistically significant, but as alkalinity actually decreases (or is similar) in some treatments (small Mercenaria, large Mercenaria control pH, small Crassostrea, large Crassostrea low pH) it would be interesting to see the relationship between these parameters and growth directly and visually.

In treatments with Ulva additions, one would expect the variability in pH to be higher due to respiratory activity and production. However, the average pH is higher but the variability in pH seems similar to treatments without Ulva. In fact, I would expect the algal-addition treatments to have a fluctuating pH and the control treatments to be stable, which could arguably have caused the differences. However the authors do not discuss this and the tables do not show these differences in variability of pH. Was the pH fluctuating on a day-night scale in the Ulva treatments? Or was the gas flowrate so high this was not discernable, and what causes the variability in the control treatments?

The nutrient and algae addition to the vessels might cause different nutrient concentrations in the treatments, with Ulva taking up nutrients while they remain suspended in the control vessels, which could have influenced results.

It is unclear what time of the year the experiments have been done (presumable summer due to hatchery times), and how the results might vary in other seasons (i.e. when Ulva is not productive).

The various sizes and the amount of different species of bivalves used in this study make it an interesting read, even though it is not entirely clear what causes the beneficial effect of the presence of Ulva (its effect on the carbonate chemistry, nutrient concentration or something else).

Specific comments:

Methods P.3, line 9 "light intensity ($\sim$200 $\mu$mol photons m-2 s-1)", how does this compare to ambient conditions?

P.3, line 23: Isochyrsis should be Isochrysis

P.4, line 17: "some estuarine environments" – representable for the environments of the study organisms and their origin?

P.4, line 32-33: "Well-pigmented, circular sections of Ulva ($\sim$3.5 cm and $\sim$7 cm for experiments in small containers and large vessels, respectively". These small containers where 1L, while the large vessels had a volume of 8L. The biomass of Ulva however, is 2x as large for the larger volume, which does not respect the ratio biomass/water volume. The authors state that the weight was consistent with the benthic coverage in Shinnecock Bay, would that mean that the 8L vessels had 2x the diameter of the small containers and would water volume not be more important than surface in this case? Or was there more than 1 disk per container (p.5., line 23 states "disks")? This section is a bit unclear.

P.5, line 16-17: "with discrete and continuous measurements of pH, dissolved oxygen, and temperature", which measurements were discrete and which continuous?

Results P.6, lines 19-20: "For the larger-sized cohort of M. mercenaria ($5.00 \pm 0.41$ mm), $\Omega$calcite and $\Omega$aragonite were significantly higher in treatments containing Ulva and significantly lower in high CO2 treatments" Throughout the manuscript's result section this way of describing the differences between high CO2 / Ulva treatments is confusing. In the highCO2+Ulva treatment the $\Omega$calcite is actually lower than the control-Ulva treatment (as expected), however from the text it appears at a first glance that all Ulva containing treatments are higher, the sentences might be clarified to prevent confusion.

Discussion Could the fact that Mytilus seems less sensitive to addition of Ulva be related to the more "natural" (no hatchery) origin of the juveniles and their exposure to environmental fluctuations vs. the more stable hatchery conditions?

If the presence of algae buffered the carbonate chemistry (p.9, line 23) and this is the mechanism for enhanced growth, this should be visible when $\Omega$calcite/aragonite is plotted vs. growth. However, the saturation state with Ulva is still considerably below 1 in the highCO2 treatments and the SD is high.

Did the authors measure nutrients at the end of the incubations? It would be interesting to explore their theory that through Ulva presence "the nitrogen assimilation effects on alkalinity outweighed the effects of photosynthetic consumption of DIC" (p.9, line 33)

Please also note the supplement to this comment:
https://www.biogeosciences-discuss.net/bg-2018-115/bg-2018-115-RC2-supplement.pdf

---

## Author Comment (AC1) · 13 Jun 2018

**RESPONSE TO REVIEWER COMMENTS *Reviewer comments in italics*; author responses to bold**

**Reviewer #1:**

Two size cohorts of hard clams, oysters, scallops, and mussels, were grown with and without macroalga Ulva in two CO2 treatments. The results show higher growth rates of bivalves in presence of Ulva, with a small benefit in the high CO2 treatment. Increased bivalve growth rates in the presence of Ulva was attributed to the increase in saturation state caused by Ulva. The study is an interesting approach to study the potential benefit of Ulva on growth of multiple bivalve species, in the context of aquaculture management with ocean acidification. The strength of this study is that the experiment was conducted on multiple species, two size classes, and there are multiple growth metrics with consistent results. The weakness of this study is the seawater chemistry and the conclusions drawn from the data. The results are intriguing and merit further exploration of why bivalves exhibited enhanced growth in the presence of Ulva. As not all factors were controlled in this experiment (e.g. unknown effect of algae and mussels on seawater chemistry, independently and by treatment), this study provides results to further develop specific hypotheses as to why these trends were observed. In its current form, I am not convinced by the conclusion that Ulva alters seawater chemistry which in turn causes increased bivalve growth under high CO2.

**We thank the reviewer for their feedback.**

1. The authors attribute what is a substantial biological response by bivalves in the presence of Ulva and high CO2 to a very MINOR increase in saturation state over time (only 0.04!). A lot of emphasis is placed on statistical comparisons of saturation state across treatments, probably because the change is so small but offers an attractive explanation. However, a statistically significant difference in a carbonate chemistry parameter across treatments does not mean that it is biologically relevant. The authors do not discuss if the magnitude of change in growth is realistic for a 0.04 change in saturation state (perhaps some summary plot showing growth metrics of each species by treatment, with aragonite saturation state of each treatment on the x-axis, would provide insight). However, Comment #2 explains why the sampling design is insufficient to characterize seawater chemistry in this experiment in the first place.

We appreciate the reviewer's perspective on this point. First, we note that small changes in saturation state, even when saturated, can be biologically important and significant. In prior studies (Talmage and Gobler, 2010, 2011), the growth of early life stage bivalves used in the present study (*Mercenaria mercenaria* and *Argopecten irradians*) was assessed under three concentrations of CO2 (pre-industrial, present-day, and elevated CO2 = 280, 390, and ~780 ppm, respectively), significant increases in growth were observed between 280 and 390 ppm CO2 which often corresponded to small changes in  $\Omega_{aragonite}$  (<0.1 units) within the saturated ranged. We note that we did refer to this example in the discussion of the manuscript. We also note that other studies have ascribed small changes in  $\Omega_{aragonite}$  to significant changes in early life stage bivalve survival (Barton et al 2012).

We agree with the reviewer that plots of saturation states against the growth would be of value hence, as suggested by the reviewer, for this revision we will provide plots showing growth metrics of each species by treatment, with aragonite saturation state of each treatment on the x-axis. We will also place the resulting statistics in tables as supplements to the manuscript, with references throughout the manuscript. To summarize these new findings, there was a strong positive and significant (p<0.05) correlation between shell length-based growth and saturation states of aragonite and calcite in all species and size classes, save for *Mytilus edulis*. In at least half of the experiments, there was a strong positive correlation and significant (p<0.05) correlation between tissue and shell weight-based growth and the saturation states of calcite and aragonite, with several additional results approaching significance (p<0.07). We are grateful for this comment by the reviewer, as it assisted in discovering these important trends.

2. The seawater chemistry sampling design and measurements are not sufficient to describe how organisms contributed to seawater chemistry or what they actually experienced.

a. Water was only sampled at the start and end of the experiment, despite multiple water changes during the closed-system experiment. If the changes in saturation state come from cumulative effect of nitrate assimilation by Ulva, this is in fact a change that since the last water change (every 3 days). It means that the bivalves mostly experienced the same saturation state across high CO2 treatments, regardless of the 0.04 change that would have occurred over 3 days.

The conclusion stated here is not supported by the data. *Ulva* is capable of the rapid uptake of nutrients, which were added after every water change. Within 24 hrs of each water change, pH values within containers with *Ulva*, regardless of CO2 concentration, were higher than in the containers without *Ulva*, meaning bivalves mostly experienced higher saturation states during experiments. We have made new plots of the day-by-day change in pH for each of our seven experiments that demonstrate the strong and significant effect of *Ulva* on pH from the start of each water change in each of the experiments. This will be included in the revision of our manuscript. We are again extremely grateful for this comment by the reviewer, as it assisted in discovering this important and what we find to be highly convincing trend.

b. Seawater chemistry was highly variable. According to the authors, Ulva changes carbonate chemistry via CO2 uptake (decreasing DIC; P9, L11-22) and/or nutrient uptake (increasing TA, estimated at 10-20 umol/kg; P9,L29). During the experiment, the effect of CO2 uptake via primary production by Ulva is presumably removed with continuous bubbling with treatment concentrations of air/CO2 gas mix (P9). However, pCO2 is quite variable across treatments and experiments, indicating that the method used for bubbling did not actually bring the system (treatments + biology) into equilibrium. For example, within one experiment, the standard error in pCO2 reported in Table S1 is up to 200 uatm (based on N=2, start and end samples?). TA also varied substantially, even across treatments without Ulva, and TA did not always increase in the presence of Ulva (Table S1, this is masked by Table 1 which somewhat deceptively summarizes treatments across all experiments). For example, TA was 230 umol/kg less in the CO2 treatment compared to control in the experiment for Mercenaria mercenaria, even without Ulva. The authors do not describe why all their measurements are so variable and inconsistent

in what they define as a well-controlled system. It is unclear if SE refers to a start and end sampling, which again is not a relevant design if the authors think that biological processes contribute to changes in seawater chemistry.

We agree with the reviewer. There was variance in the chemistry driven by the biology and that the productivity of the *Ulva* was capable of overpowering the bubbling. We note experimental that the variability reported is based on replicate vessels with n=4 of the final time point and the daily measurement of pH, not two time points. We will clarify this point in the revision. Importantly, these are true biological replicates representing the cumulative effects of the whole experimental ecosystem: *Ulva*, bivalves, microbial communities, etc on the chemistry. While there is variance generated by these communities, the presence of *Ulva* indeed caused the calcium carbonate saturation states within experimental vessels to be statistically significantly higher than vessels without them and caused a day-by-day rise in pH which we will provide in the revised version of the manuscript, along with the new statistically significant regressions of bivalve growth with saturation state.

We are uncertain as to why the reviewer would suggest we were being 'deceptive' by creating summary supplementary tables. If the tables were available to the reviewer and will be available to all readers, we find this to be full transparent. We note, however, while abiotic systems bubbled with CO2 generally have consistent alkalinity, alkalinity can be affected by multiple biotic and abiotic processes associated all living organisms within each experimental vessel: Respiration, photosynthesis, shell dissolution, calcification, nitrate uptake, phosphate uptake, ammonium uptake, microbial degradation, etc.

c. Chemistry was calculated using pH that was measured by a Durafet but no information on calibration and quality control was provided. It is unclear how and where the daily pH measurements are used.

The DuraFET III used in the present study were calibrated with a seawater pH standard, as per Dickson (1993), and compared to measurements made spectrophotometrically using *m*-cresol (Dickson et al., 2007). Both methods yielded pH measurements that were identical and never significantly different. We agree with the reviewer's concerns here and will add this information to the revised manuscript. Measurements of pH were used in the calculation of carbonate chemistry, which is stated on P4, L11-12. We will also be providing the day-by-day pH values from experiments to illustrate the effects of nitrate uptake by *Ulva*.

d. For all of the above reasons, I am not convinced that photosynthesis or nitrate assimilation by Ulva increased saturation state which then enhanced growth of bivalves (as claimed on P11, L29-30). Unless the authors can clarify these points, alternative hypotheses should be discussed. For example, could proliferation of algal cells in high CO2 have provided more food to the bivalves and therefore contributed to their growth.

We applaud the reviewer's skepticism as this is a core element of the review process. As the reviewer requested, we have shown that the growth rates of bivalves are significantly correlated with the saturation states of two forms of calcium carbonate and that calcium carbonate saturation states were always significantly higher within the treatments with *Ulva*, two key data sets supporting the hypothesis that improved conditions for calcification was the key factor driving trends observed in this study. The reviewer has provided an alternative hypothesis but one that does not fit the data since if high CO2 led to the proliferation of algal cells, and thus more food for the bivalves, once would expect their growth to increase but our results showed they actually decreased growth under elevated CO2. We do, however, agree with the reviewer's point that differences in algal cells within treatments could impact the growth of bivalves. Therefore, for this revision we have enumerated final algal cell densities within experimental vessels for all treatments. To summarize these findings, there were no significant differences in algal cell counts across any treatment within individual experiments. A table with this data will be created for this revision and added to the supplementary materials.

**After all, nutrients were added and this would benefit Isochrysis spp. (spelling error on P3,L23) and Chaetoceros spp.**

Yes, nutrients were added to all experiments and vessels. This point is specified on P3, L17. Despite the plausibility that the microalgae could have influenced the growth of bivalves, analyses of phytoplankton cell densities within each treatment and experiment rule out this possibility as there were no difference in algal cell counts across treatments within individual experiments. A table has been created and will be added to the supplementary materials.

3. The extensive discussion (e.g. last four paragraphs) on macroalgae/seagrass benefits to bivalves detracts from the discussion of the results of this study, and makes the authors appear biased towards the hypothesis that macroalgae will mitigate ocean acidification (e.g., their interpretation of Unsworth et al 2012 on P11,L17, comments below). The ability for seagrass and macroalgae to chemically buffer ocean acidification (e.g., P12, L1-2) is not a fact, and needs to be considered in the context of the greater coastal environment that the habitat is in (e.g., freshwater inputs, upwelling, water residence time, etc., e.g. see Cyronak at al 2018 "Short-term spatial and temporal carbonate chemistry variability in two contrasting seagrass meadows: implications for pH buffering capacities"). The authors do not discuss the fact that their experiment was conducted in a closed system. It is unrealistic to conclude that a minute impact on alkalinity by Ulva (if verified, see comment 1 & 2) would mitigate ocean acidification in an open system. For these reasons, extrapolating these results to field applications should not take up more than a paragraph, and the authors should only do so if all of the issues with seawater chemistry can be sufficiently resolved.

We do not suggest that macroalgae alone can mitigate ocean acidification, but rather merely that primary productivity and/or nitrate assimilation by macroalgae may provide a small temporal and spatial refuge for bivalves and other calcifying organisms as has been stated and concluded in prior studies. Given the scale, this may be particularly relevant to bivalves in an aquaculture setting with macroalgae purposely grown in copious quantities in close proximity to bivalves potentially providing regional "chemical resilience". As per the reviewer's comments, we will, however, significantly scale back this discussion. Title: based on the issues with seawater chemistry, this title may need to be revised

We believe the title aptly describes the paper given the linear relationships between saturation states and the growth of bivalves and the day-by-day increases in pH provided by *Ulva*.

Abstract: remove p-values

**We will remove the p-values from the Abstract.**

- Half of this study has to do with large vs. small bivalves but the significance of this is not mentioned in the introduction. Please add the motivation for this in the Introduction.

This was done since vulnerability of bivalves to acidification can be size-dependent. We will add this information to the Introduction.

- P2, L18: specify that pH and saturation state in seagrass meadows provide \*temporal\* refuge from acidification (as pH also declines below background seawater pH at night or in winter seasons).

We agree with the reviewer and have specified that daytime primary productivity increases pH and saturation states of aragonite, which provides a temporal refuge from acidification.

- Were nutrients added to vessels without Ulva as well? If not, the presence of Ulva is confounded with presence of nutrients which could influence the growth of Isochrysis and Chaetoceros and therefore the food supply by treatment.

Nutrients were added to all experimental vessels, *Ulva* or not, for the reason that the reviewer states. This point is specified on P3, L17.

- P3, L 24: how can 'ad libitum' food supply be exactly 4 x 104 cells mL-1 d-1?

For the bivalves used in the present study, the rate of 4x104 cells mL-1 d-1 of the specified microalgae is an amount that is more than sufficient (*'ad libitum'*) for the growth of the studied bivalves as per Helm MM, Bourne N, Lovatelli A (2004). Hatchery Culture of Bivalves: A Practical Manual. Rome, Italy: Food and Agriculture Organization of the United Nations (FAO), which we will reference in our revision. In addition, we will demonstrate in this revision that there were always excess algal cells at the end of experiments, providing direct evidence that this food supply was indeed, *ad libitum*.

- *Report on assumptions of ANOVA (i.e., do residuals exhibit a normal distribution? was this tested?)*

In this revision we will report on the assumptions of the ANOVA tests. In order to ensure that our data met the assumptions of the ANOVA (normality and equal variance), all data

**were log transformed before ANOVA were performed. We will add these details to the Methods section and have update the supplementary materials to reflect these changes.**

- P4, L34: add # of circles of algae added to each vessel. Was this scaled by container volume for small (1 L) and large bivalves (8 L)? If Ulva changes seawater chemistry in a consistent way, this data can be used to explore that (e.g., weight to volume and magnitude change in seawater chemistry).

A single disk of *Ulva* was added per container, which we will include in the Methods. In terms of weight, the amount of *Ulva* added to 1 L and 8 L containers was consistent with the benthic coverage of *Ulva* in Shinnecock Bay based on several years of benthic trawl data as well as other estuarine regions (Liu et al., 2015; Sfriso et al., 2001) and thus, yes, it was scaled to the size of the vessels. This point is specified in the Methods on P4, L31-34 and P5, L1-5.

- Tables in supplement: check consistency of \* with p < 0.05

**We will change the text within the supplementary materials to make consistent use of asterisks for significant results.**

- Please report the actual p-values in the text since the tables are in supplemental files.

**We will change the text to reflect the actual p-values within the Results section.**

- I don't understand how ANOVA results are used to make statements like "When in the presence of Ulva, shell length-based growth was significantly increased by 42% (Two-way ANOVA; p<0.05)" when it is unlikely that the % change is the same in high CO2 and low CO2 treatments. If the authors are reporting the effect of Ulva only at high CO2, then the statistics should come from the Tukey post-hoc comparison. Authors should also report on the interaction of the two-way ANOVA (significant or not).

In this text, we reference Fig. 3 which demonstrates the increased shell length-based growth in the presence of *Ulva* (by 42%). The reference to Table S4 shows the ANOVA results, not the percent difference. The point being illustrated here is that growth increased in the presence of *Ulva* by a certain percent, which, by way of Two-way ANOVA, was found to be statistically significant. We agree that with the reviewer that the 42% increase may not be the same within elevated and ambient CO2 treatments, and will change the text to separate the percent increase between the two CO2 treatments for the revised version of the manusctipt.

- I was expecting the Ulva results in the Results section. It's not critical, but a small point of confusion.

We had to not include *Ulva* growth results in the Results section since it is not related to the primary goals of the study and since our previous studies have already reported on

enhanced growth in *Ulva* incubated under elevated CO2. We will add the mean response of *Ulva* as a supplementary figure for this revision and refer to this at the end of the results.

- P5, L35: report tests of ANOVA assumptions, report p-values that are corrected for multiple comparisons.

For this revision we will specifically report on the use of Shapiro-Wilk test to test for normality, in addition to an equal variance test, both of which are built into SigmaPlot. We performed log transformations of the data to ensure that they passed both tests and will update the supplementary materials to reflect this change. We will also change the text within the manuscript to state what assumptions were made for ANOVA.

- P11,L16-19: this statement is incorrect. Unsworth et al 2012 is a theoretical modeling study. Model results were then applied to coral calcification rates that came from laboratory-based experiments. The authors themselves state that the results from the modeling need to be field tested.

We thank the reviewer for pointing out our error. We will change the text to state that Unsworth et al. (2012) used a theoretical model to determine that coral calcification downstream from seagrass meadows could increase by ~18% should the full extent of a seagrass meadow's ability to increase pH and  $\Omega_{aragonite}$  be realized in a natural setting.

- Discussion should include information about the magnitude of the beneficial effect of Ulva under high CO2.

For this revision, we will state the percent increase in growth rate of the bivalves in the discussion.

Table 1: indicate which parameters were measured, and sample size (N).

We agree with including the sample size, and will add an asterisk next to the parameters that were measured but not the ones that were calculated and explain what the asterisk represents in the table legend.

*Figures: define error bars and indicate when there are significant differences among groups*

We have indicated the definition of the error bars within the figure captions, and have placed significant differences within the figures, specifically the main treatment effects  $(CO_2 \text{ and } Ulva)$ .

---

## Author Comment (AC2) · 13 Jun 2018

RESPONSE TO REVIEWER COMMENTS
*Reviewer comments in italics*; **author responses to bold**

**Reviewer #2:**

*"The ability of macroalgae to mitigate the negative effects of ocean acidification on four species of North Atlantic bivalve" This paper evaluates the effect of the presence of the macroalga Ulva rigida on the growth of four North Atlantic bivalve species, Mercenaria mercenaria, Crassostrea virginica, Argopecten irradians and Mytilus edulis. The authors have used small and larger sizes of three out of four species, specifically the three obtained from hatcheries. The pCO2 levels the bivalves are exposed to are high, but conceivable for estuarine systems. The authors claim that "saturation states for calcium carbonate (Ω) were significantly higher in the presence of Ulva under both ambient and elevated CO2 delivery rates (p<0.05)", and that "alkalinity was increased by the presence of Ulva". This might be statistically significant, but as alkalinity actually decreases (or is similar) in some treatments (small Mercenaria, large Mercenaria control pH, small Crassostrea, large Crassostrea low pH) it would be interesting to see the relationship between these parameters and growth directly and visually.*

**We agree with the reviewer's assessment. For this revision we will provide regression analyses of saturation states for calcium carbonate with the growth rates of bivalves to demonstrate the specific relationship between these two parameters. These analyses will appear as supplementary tables and will be discussed in the results and discussion. To summarize these new findings, there was a strong positive and significant ($p<0.05$) correlation between shell length-based growth and saturation states of aragonite and calcite in all species and size classes, save for *Mytilus edulis*.**

**We note that alkalinity is affected by many processes and while nitrate uptake will increase alkalinity, other processes may decrease it and that prior research has definitively demonstrated that saturation states for calcium carbonate are the key factor dictating the effects of acidification on bivalves.**

*In treatments with Ulva additions, one would expect the variability in pH to be higher due to respiratory activity and production. However, the average pH is higher but the variability in pH seems similar to treatments without Ulva. In fact, I would expect the algal-addition treatments to have a fluctuating pH and the control treatments to be stable, which could arguably have caused the differences. However the authors do not discuss this and the tables do not show these differences in variability of pH. Was the pH fluctuating on a day-night scale in the Ulva treatments? Or was the gas flowrate so high this was not discernable, and what causes the variability in the control treatments?*

**The reviewer is correct that the treatments with *Ulva* had more variability in pH but did, on average, have higher pH levels. For this revision, we have made plots showing the changes in pH over time for the *Ulva* treatments to demonstrate that there is variability but that the pH rose in these treatments after each water change in every experiment, likely due to the uptake of nitrate and the assimilation of $CO_2$ by *Ulva*.**

*The nutrient and algae addition to the vessels might cause different nutrient concentrations in the treatments, with Ulva taking up nutrients while they remain suspended in the control vessels, which could have influenced results.*

**We agree with the reviewer that additions of nutrients and algae might cause different nutrient concentrations within treatments, and that *Ulva* may alter nutrient concentrations. If we agree that more nutrients could increase algae concentrations, this could result in more growth. However, we note that in treatments without Ulva where there might be more nutrients, bivalve growth rates were lower, not higher as would be predicted. More importantly, for the revision we will provide the newly obtained data on phytoplankton cell counts which were found to be highly similar and not statistically different across all experimental treatments in each experiment. Finally, but importantly, we note that any differences in nutrients among the vessels would occur in an ecosystem setting as well with more nitrate assimilation and removal and thus an increase in alkalinity in times and places where there is more *Ulva*. Hence, any differences on this front would be realistic in an ecosystem setting.**

*It is unclear what time of the year the experiments have been done (presumable summer due to hatchery times), and how the results might vary in other seasons (i.e. when Ulva is not productive).*

**The reviewer's presumption is correct as the experiments occurred throughout summer 2017, which is the peak growing season for bivalves and *Ulva*. We targeted this season specifically for that reason, although it should be noted that within the collection site of *Ulva*, the macroalgae appear during the early days of spring, and persists into the end of the fall months, which is beyond the time that our experiments were concluded. During period when *Ulva* grows more slowly (spring and fall) it would be expected that its growth would be slower and its ability to mitigate acidification would be lower.**

*The various sizes and the amount of different species of bivalves used in this study make it an interesting read, even though it is not entirely clear what causes the beneficial effect of the presence of Ulva (its effect on the carbonate chemistry, nutrient concentration or something else).*

**We agree with the reviewer that the exact cause of the increased bivalve growth in the presence of *Ulva* cannot be exclusively tied to a singular cause. However, we believe the new data generated for our revision make the case for the mitigation of acidification even stronger. We believe the new regression analyses we will include that depicts the significant linear relationships between calcium carbonate saturation states and bivalve growth makes the carbonate chemistry angle more convincing. We believe the day-by-day decreases in pH provided by Ulva during experiment makes the carbonate chemistry angle more convincing. Finally, our inclusion of phytoplankton density data showing there are no differences among treatments indicate this was not a driver of the findings.**

*Specific comments: Methods P.3, line 9 "light intensity ($\sim$200 µmol photons m-2 s-1)", how does this compare to ambient conditions?*

**Light intensity used in all experiments was set to mimic ambient light intensity where *Ulva* grows in near shore regions. We will add this information to the manuscript to specify this.**

*P.3, line 23: Isochyrsis should be Isochrysis*

**We will make the suggested change.**

*P.4, line 17: "some estuarine environments" – representable for the environments of the study organisms and their origin?*

**Yes. For example, Wallace et al. (2014) observed $pCO_2$ concentrations exceeding 2,000 µatm in Jamaica Bay, NY, USA, which hosts the bivalve and macroalgae species used in the present study.**

*P.4, line 32-33: "Well-pigmented, circular sections of Ulva (~3.5 cm and ~7 cm for experiments in small containers and large vessels, respectively". These small containers where 1L, while the large vessels had a volume of 8L. The biomass of Ulva however, is 2x as large for the larger volume, which does not respect the ratio biomass/water volume. The authors state that the weight was consistent with the benthic coverage in Shinnecock Bay, would that mean that the 8L vessels had 2x the diameter of the small containers and would water volume not be more important than surface in this case? Or was there more than 1 disk per container (p.5., line 23 states "disks")? This section is a bit unclear.*

**The amount of *Ulva* used was based on tissue weight, not tissue surface area, and the amount of *Ulva* added to 1 L and 8 L containers was consistent with the benthic coverage of *Ulva* in Shinnecock Bay based on several years of benthic trawl data as well as other estuarine regions (Liu et al., 2015; Sfriso et al., 2001). This point is specified in the Methods on P4, L31-34 and P5, L1-5. Considering the 2-dimensional nature in which interactions of the bivalves and the macroalgae would occur, it would make more sense to base the amount of macroalgae used on the surface area of the container, and not necessarily the volume.**

*P.5, line 16-17: "with discrete and continuous measurements of pH, dissolved oxygen, and temperature", which measurements were discrete and which continuous?*

**We measured pH and temperature discretely and dissolved oxygen continuously. We will change the text to specify this difference.**

*Results P.6, lines 19-20: "For the larger-sized cohort of M. mercenaria (5.00 ± 0.41 mm), Ωcalcite and Ωaragonite were significantly higher in treatments containing Ulva and significantly lower in high CO2 treatments" Throughout the manuscript's result section this way of describing the differences between high CO2 / Ulva treatments is confusing. In the highCO2+Ulva treatment the Ωcalcite is actually lower than the control-Ulva treatment (as expected), however from the text it appears at a first glance that all Ulva containing treatments are higher, the sentences might be clarified to prevent confusion.*

We intended to specify that $\Omega_{calcite}$ and $\Omega_{aragonite}$, although significantly lower under elevated $CO_2$ concentrations in general, were significantly higher in the presence of *Ulva* in both ambient and elevated $CO_2$ treatments. We agree with the reviewer that the sentence structure used throughout the manuscript may cause confusion and will change the text to separate any significant differences in $\Omega_{calcite}$ and $\Omega_{aragonite}$, be it under elevated $CO_2$ conditions, or in the presence of *Ulva*. We will also include references to the respective figures that show $\Omega_{calcite}$ and $\Omega_{aragonite}$, which would make it clear that $\Omega_{calcite}$ and $\Omega_{aragonite}$ are lower under elevated $CO_2$, but higher in the presence of *Ulva* in both ambient and elevated $CO_2$ treatments.

*Discussion Could the fact that Mytilus seems less sensitive to addition of Ulva be related to the more "natural" (no hatchery) origin of the juveniles and their exposure to environmental fluctuations vs. the more stable hatchery conditions?*

This is a good point raised by the reviewer. The area within Shinnecock Bay where *Mytilus* were collected is well-flushed and not prone to significant decreases in dissolved oxygen or pH or increases in $pCO_2$. In addition, *Mercenaria* and *Argopecten* within the hatchery at Stony Brook University in Southampton are exposed to similar environmental conditions that are found in the collection sites in Shinnecock Bay from which these original broodstock came. We will clarify the recent origin of the broodstock used in experiments in the methods.

*If the presence of algae buffered the carbonate chemistry (p.9, line 23) and this is the mechanism for enhanced growth, this should be visible when $\Omega$calcite/aragonite is plotted vs. growth. However, the saturation state with Ulva is still considerably below 1 in the highCO2 treatments and the SD is high.*

This was an excellent suggestion by the reviewer and for this revision, we have now included regression of $\Omega_{calcite}$ and $\Omega_{aragonite}$ vs. growth which in nearly all cases provided significant correlations. While the $\Omega$ is below 1 in many high CO2 cases, prior studied have shown early life stage bivalves do grow, albeit slower, under such conditions (Talmage and Gobler 2010, 2011).

*Did the authors measure nutrients at the end of the incubations? It would be interesting to explore their theory that through Ulva presence "the nitrogen assimilation effects on alkalinity outweighed the effects of photosynthetic consumption of DIC" (p.9, line 33)*

No, nutrient concentrations were not measured at the beginning or the end of experiments. Due to the multiple water changes that occurred throughout experiments, measuring only the final nutrient concentrations would not accurately represent actual nutrient concentrations throughout.

---

## Author Response (AR1)

RESPONSE TO REVIEWER COMMENTS
*Reviewer comments in italics*; **author responses to bold**

***Reviewer #1:***

*Two size cohorts of hard clams, oysters, scallops, and mussels, were grown with and without macroalga Ulva in two CO2 treatments. The results show higher growth rates of bivalves in presence of Ulva, with a small benefit in the high CO2 treatment. Increased bivalve growth rates in the presence of Ulva was attributed to the increase in saturation state caused by Ulva.*

*The study is an interesting approach to study the potential benefit of Ulva on growth of multiple bivalve species, in the context of aquaculture management with ocean acidification. The strength of this study is that the experiment was conducted on multiple species, two size classes, and there are multiple growth metrics with consistent results. The weakness of this study is the seawater chemistry and the conclusions drawn from the data. The results are intriguing and merit further exploration of why bivalves exhibited enhanced growth in the presence of Ulva. As not all factors were controlled in this experiment (e.g. unknown effect of algae and mussels on seawater chemistry, independently and by treatment), this study provides results to further develop specific hypotheses as to why these trends were observed. In its current form, I am not convinced by the conclusion that Ulva alters seawater chemistry which in turn causes increased bivalve growth under high CO2.*

**We thank the reviewer for their feedback.**

*1. The authors attribute what is a substantial biological response by bivalves in the presence of Ulva and high CO2 to a very MINOR increase in saturation state over time (only 0.04!). A lot of emphasis is placed on statistical comparisons of saturation state across treatments, probably because the change is so small but offers an attractive explanation. However, a statistically significant difference in a carbonate chemistry parameter across treatments does not mean that it is biologically relevant. The authors do not discuss if the magnitude of change in growth is realistic for a 0.04 change in saturation state (perhaps some summary plot showing growth metrics of each species by treatment, with aragonite saturation state of each treatment on the x-axis, would provide insight). However, Comment #2 explains why the sampling design is insufficient to characterize seawater chemistry in this experiment in the first place.*

**We appreciate the reviewer's perspective on this point. First, we note that small changes in saturation state, even when saturated, can be biologically important and significant. In prior studies Barton et al (2012, Limnol, Oceanogr) saw that survival of early life stage Pacific oysters were correlated with $\Omega_{aragonite}$ even in the saturated range of values and that small change made substantial differences. Similarly, the growth of early life stage bivalves used in the present study (*Mercenaria mercenaria* and *Argopecten irradians*) was assessed under three concentrations of $CO_2$ (280, 390, and ~780 ppm) and significant differences in growth were observed between 280 and 390 ppm $CO_2$ which often corresponded to small changes in $\Omega_{aragonite}$ (<0.1 units) within the**

**saturated ranged (Talmage and Gobler, 2010 PNAS, 2011, PLOS One). We note that we did refer to these examples in the discussion of the manuscript.**

**We agree with the reviewer's suggestion that plots of saturation states against the growth would be important to examine. Therefore, as suggested by the reviewer, for this revision we have made plots for every experiment showing growth rates of each species as a function of aragonite and calcite saturation state for each treatment on the x-axis. We have placed the resulting regression statistics in tables as supplements to the manuscript (new Table S10), with references to the table throughout the manuscript. To summarize these findings, there were strong positive and significant ($p<0.05$) correlations between shell length-based growth and saturation states of aragonite and calcite for all species and size classes, save for the single *Mytilus edulis* experiment. In at least half of the experiments, there was a strong positive correlation and significant ($p<0.05$) correlation between tissue and shell weight-based growth and the saturation states of calcite and aragonite, with several additional results approaching significance ($p<0.07$).**

*2. The seawater chemistry sampling design and measurements are not sufficient to describe how organisms contributed to seawater chemistry or what they actually experienced.*

*a. Water was only sampled at the start and end of the experiment, despite multiple water changes during the closed-system experiment. If the changes in saturation state come from cumulative effect of nitrate assimilation by Ulva, this is in fact a change that since the last water change (every 3 days). It means that the bivalves mostly experienced the same saturation state across high CO2 treatments, regardless of the 0.04 change that would have occurred over 3 days.*

**We thank the review for this comment, as it motivated us to dig deeper into the data we had already collected to discover that, in fact, the bivalves mostly experienced different saturation states within the *Ulva* treatments across the experiments. *Ulva* is capable of the rapid uptake of nutrients, which were added after every water change. Within 24 hr of each water change, pH values within containers with *Ulva*, regardless of CO₂ concentration, were higher than in the containers without *Ulva*, meaning bivalves mostly experienced higher saturation states during experiments. We have provided plots to show this and now make reference to these new plots within the manuscript (Figures S2-S3).**

*b. Seawater chemistry was highly variable. According to the authors, Ulva changes carbonate chemistry via CO2 uptake (decreasing DIC; P9, L11-22) and/or nutrient uptake (increasing TA, estimated at 10-20 umol/kg; P9,L29). During the experiment, the effect of CO2 uptake via primary production by Ulva is presumably removed with continuous bubbling with treatment concentrations of air/CO2 gas mix (P9). However, pCO2 is quite variable across treatments and experiments, indicating that the method used for bubbling did not actually bring the system (treatments + biology) into equilibrium. For example, within one experiment, the standard error in pCO2 reported in Table S1 is up to 200 uatm (based on N=2, start and end samples?). TA also varied substantially, even across treatments without Ulva, and TA did not always increase in the presence of Ulva (Table S1, this is*

*masked by Table 1 which somewhat deceptively summarizes treatments across all experiments). For example, TA was 230 umol/kg less in the CO2 treatment compared to control in the experiment for Mercenaria mercenaria, even without Ulva. The authors do not describe why all their measurements are so variable and inconsistent in what they define as a well-controlled system. It is unclear if SE*
5   *refers to a start and end sampling, which again is not a relevant design if the authors think that biological processes contribute to changes in seawater chemistry.*

**We agree with the reviewer. There was variance in the chemistry during our experiments. Prior to starting any experiment, out vessels filled with seawater only were bubbled at a constant rate**
10   **which created a very stable system at full equilibrium for many days. Once biological organisms were introduced, however, as the reviewer correctly stated, the individual vessels were no longer in a simple abiotic equilibrium, but rather represented dynamic ecosystems with full complement of living organisms (*Ulva*, bivalves, phytoplankton added as food, microbial communities) undergoing all of the biological processes that influence carbonate chemistry (uptake and release**
15   **of ions, shell formation, shell dissolution, etc). One would not expect such systems to be in any kind of equilibrium. Furthermore, they each represented true biological replicates with a different set of bivalves, a different fronds of *Ulva*, different phytoplankton cells added and our variance in our reporting of our carbonate chemistry was based on replicate vessels (i.e. ecosystems) with n=4. Thus, these were true biological replicates representing the cumulative**
20   **effects of the whole experimental ecosystem (i.e. *Ulva*, bivalves, phytoplankton added as food, microbial communities, uptake and release of ions, etc) on the chemistry. While there is variance generated by these communities, the presence of *Ulva* consistently caused the calcium carbonate saturation states within experimental vessels to be statistically significantly higher than vessels without them. In addition, in this revision, we demonstrate in the new Figures S2-S3, pH changes**
25   **occurred with 24 of each water change, showing that, indeed, the chemistry was dynamic and varied in these living, biological systems, but that the presence of *Ulva* had a rapid and discernable effect on pH and carbonate chemistry.**

**We are uncertain as to how we were being 'deceptive' by creating both summary supplementary**
30   **tables given they are presenting all of our data making it available for everyone to read. While abiotic systems bubbled with $CO_2$ generally have consistent alkalinity, alkalinity can be affected by multiple biotic and abiotic processes associated all living organisms within each experimental vessel: Respiration, photosynthesis, shell dissolution, calcification, nitrate uptake, phosphate uptake, ammonium uptake, microbial degradation, etc.**

*c. Chemistry was calculated using pH that was measured by a Durafet but no information on calibration and quality control was provided. It is unclear how and where the daily pH measurements are used.*

40   **The DuraFET III used in the present study were calibrated with a seawater pH standard, as per Dickson (1993), and compared to measurements made spectrophotometrically using *m*-cresol (Dickson et al., 2007). Both methods yielded pH measurements that were identical and never**

**significantly different. We agree with the reviewer's concerns here and have added this information to the manuscript, which is now stated on P4, L3-7. Measurements of pH were used in the calculation of carbonate chemistry, which is stated on P4, L15-17. We have also provided the day-by-day pH values from experiments to illustrate the effects of nitrate uptake by *Ulva***

5 **(Figures S2-S3).**

*d. For all of the above reasons, I am not convinced that photosynthesis or nitrate assimilation by Ulva increased saturation state which then enhanced growth of bivalves (as claimed on P11, L29-30). Unless the authors can clarify these points, alternative hypotheses should be discussed. For example, could*

10 *proliferation of algal cells in high CO2 have provided more food to the bivalves and therefore contributed to their growth.*

**We applaud the reviewer's skepticism as this is a core element of the review process and it forced us to generate additional plots and analyses of our experiments that provided evidence that is**

15 **substantially more robust than our original submission. As the reviewer requested, we have shown that the growth rates of bivalves are significantly correlated with the saturation states of two forms of calcium carbonate (new Table S10) and that calcium carbonate saturation states were always significantly higher within the treatments with *Ulva*, two key data sets supporting the hypothesis that improved conditions for calcification was the key factor driving trends observed**

20 **in this study. The reviewer has provided an alternative hypothesis but one that must be rejected given the data from our study. If high $CO_2$ led to the proliferation of algal cells, and thus more food for the bivalves, one would expect bivalve growth to increase. However, our results showed bivalve growth actually decreased growth under elevated $CO_2$. That being said, we agree with the reviewer's point that differences in algal cells within treatments could impact the growth of**

25 **bivalves. Therefore, following the reviewer's line of reasoning, for this revision we have enumerated final algal cell densities within experimental vessels for all treatments. To summarize these findings, there were no significant differences in microalgal cell counts across any treatment within individual experiments. In addition to updating Tables S1 to show these results, a new table with this data has been created for this revision and added to the supplementary materials**

30 **(Tables S13).**

*After all, nutrients were added and this would benefit Isochrysis spp. (spelling error on P3,L23) and Chaetoceros spp.*

35 **Yes, nutrients were added to all experiments and vessels. This point is specified on P3, L20-22. Despite the plausibility that the microalgae could have influenced the growth of bivalves, analyses of phytoplankton cell densities within each treatment and experiment rule out this possibility as there were no significant difference in algal cell counts across treatments within individual experiments. Table S1 has been updated to show these results and a table has been created and**

40 **will be added to the supplementary materials (Table S13).**

*3. The extensive discussion (e.g. last four paragraphs) on macroalgae/seagrass benefits to bivalves detracts from the discussion of the results of this study, and makes the authors appear biased towards the hypothesis that macroalgae will mitigate ocean acidification (e.g., their interpretation of Unsworth et al 2012 on P11,L17, comments below). The ability for seagrass and macroalgae to chemically buffer ocean acidification (e.g., P12, L1-2) is not a fact, and needs to be considered in the context of the greater coastal environment that the habitat is in (e.g., freshwater inputs, upwelling, water residence time, etc., e.g. see Cyronak at al 2018 "Short-term spatial and temporal carbonate chemistry variability in two contrasting seagrass meadows: implications for pH buffering capacities"). The authors do not discuss the fact that their experiment was conducted in a closed system. It is unrealistic to conclude that a minute impact on alkalinity by Ulva (if verified, see comment 1 & 2) would mitigate ocean acidification in an open system. For these reasons, extrapolating these results to field applications should not take up more than a paragraph, and the authors should only do so if all of the issues with seawater chemistry can be sufficiently resolved.*

**We do not believe or suggest that macroalgae alone can mitigate ocean acidification, but merely that primary productivity and/or nitrate assimilation by macroalgae may provide a temporal and/or spatial refuge for bivalves and other calcifying organisms as has been stated and concluded in prior studies. Given the scale, this may be particularly relevant to bivalves in an aquaculture setting with macroalgae grown in copious quantities in close proximity to bivalves potentially providing a "chemical resilience". As per the reviewer's comments, we have significantly scaled back this discussion for the revised version of this mnuscript.**

*Title: based on the issues with seawater chemistry, this title may need to be revised*

**We have believe the title aptly describes the paper given the linear relationships between saturation states and the growth of bivalves and the < 24 h increases in pH associated with the presence of *Ulva* during experiments.**

*Abstract: remove p-values*

**We have removed the p-values from the Abstract.**

*- Half of this study has to do with large vs. small bivalves but the significance of this is not mentioned in the introduction. Please add the motivation for this in the Introduction.*

**This was done since vulnerability of bivalves to acidification can be size-dependent. We have added this information to the Introduction.**

*- P2, L18: specify that pH and saturation state in seagrass meadows provide \*temporal\* refuge from acidification (as pH also declines below background seawater pH at night or in winter seasons).*

**We agree with the reviewer and have specified that daytime primary productivity increases pH and saturation states of aragonite, which provides a temporal refuge from acidification.**

*- Were nutrients added to vessels without Ulva as well? If not, the presence of Ulva is confounded with presence of nutrients which could influence the growth of Isochrysis and Chaetoceros and therefore the food supply by treatment.*

**Nutrients were added to all experiments, *Ulva* or not, for the reason that the reviewer states. This point is specified on P3, L20-22.**

*- P3, L 24: how can 'ad libitum' food supply be exactly 4 x 104 cells mL-1 d-1?*

**For the bivalves used in the present study, the rate of $4 \times 10^4$ cells mL$^{-1}$ d$^{-1}$ of the specified microalgae is an amount that is more than sufficient for the growth of said bivalves, regardless of size as per Helm MM, Bourne N, Lovatelli A (2004). Hatchery Culture of Bivalves: A Practical Manual. Rome, Italy: Food and Agriculture Organization of the United Nations (FAO), which we reference in this revision. In addition, we have demonstrated in this revision that there were always excess algal cells at the end of experiments (more than the amount present on day one), providing the direct evidence that this was indeed, *ad libitum* feeding.**

*- Report on assumptions of ANOVA (i.e., do residuals exhibit a normal distribution? was this tested?)*

**In this revision we have reported on the assumptions of the ANOVA tests. In order to ensure that our data met the assumptions of the ANOVA (normality and equal variance), all data were log transformed before ANOVA were performed. We have added these details to the Methods section and have update the supplementary materials to reflect these changes.**

*- P4, L34: add # of circles of algae added to each vessel. Was this scaled by container volume for small (1 L) and large bivalves (8 L)? If Ulva changes seawater chemistry in a consistent way, this data can be used to explore that (e.g., weight to volume and magnitude change in seawater chemistry).*

**A single disk of *Ulva* was added per container, which we include in the Methods. In terms of weight, the amount of *Ulva* added to 1 L and 8 L containers was consistent with the benthic coverage of *Ulva* in Shinnecock Bay based on several years of benthic trawl data as well as other estuarine regions (Liu et al., 2015; Sfriso et al., 2001) and thus, yes, it was scaled to the size of the vessels. This point is specified in the Methods on P5, L4-8.**

*- Tables in supplement: check consistency of \* with p<0.05*

**We have changed the text within the supplementary materials to make consistent use of asterisks for significant results.**

- Please report the actual p-values in the text since the tables are in supplemental files.

**We have changed the text to reflect the actual p-values within the Results section.**

5 *- I don't understand how ANOVA results are used to make statements like "When in the presence of Ulva, shell length-based growth was significantly increased by 42% (Two-way ANOVA; p<0.05)" when it is unlikely that the % change is the same in high CO2 and low CO2 treatments. If the authors are reporting the effect of Ulva only at high CO2, then the statistics should come from the Tukey post-hoc comparison. Authors should also report on the interaction of the two-way ANOVA (significant or not).*

**In this text, we reference Fig. 3 which demonstrates the increased shell length-based growth in the presence of *Ulva* (by 42%). The reference to Table S4 shows the ANOVA results, not the percent difference. The point being illustrated here is that growth increased in the presence of *Ulva* by a certain percent, which, by way of Two-way ANOVA, was found to be significant. We agree that**
15 **with the reviewer that the 42% increase may not be the same within elevated and ambient $CO_2$ treatments, and have changed the text to separate the percent increase between the two $CO_2$ treatments.**

*- I was expecting the Ulva results in the Results section. It's not critical, but a small point of confusion.*

**We had to not include *Ulva* growth results in the Results section since it is not related to the primary goals of the study and since our previous studies have already reported on enhanced growth in *Ulva* incubated under elevated $CO_2$. We have added the mean response of *Ulva* as a supplementary figure for this revision and refer to this at the end of the results.**
25
*- P5, L35: report tests of ANOVA assumptions, report p-values that are corrected for multiple comparisons.*

**For this revision we have specifically reported on the use of Shapiro-Wilk test to test for**
30 **normality, in addition to an equal variance test, both of which are built into SigmaPlot. We performed log transformations of the data to ensure that they passed both tests and will update the supplementary materials to reflect this change. We have also changed the text within the manuscript to state what assumptions were made for ANOVA.**

35 *- P11,L16-19: this statement is incorrect. Unsworth et al 2012 is a theoretical modeling study. Model results were then applied to coral calcification rates that came from laboratory-based experiments. The authors themselves state that the results from the modeling need to be field tested.*

**We thank the reviewer for pointing this out. We have removed the reference to Unsworth et al.**
40 **(2012).**

*- Discussion should include information about the magnitude of the beneficial effect of Ulva under high CO2.*

**We agree with the reviewer. For this revision, we state the percent increase in growth rate of the bivalves in the discussion.**

*Table 1: indicate which parameters were measured, and sample size (N).*

**We agree with including the sample size and have added an asterisk next to the parameters that were measured but not the ones that were calculated and explain what the asterisk represents in the table legend.**

*Figures: define error bars and indicate when there are significant differences among groups*

**We have indicated the definition of the error bars within the figure captions, and have placed significant differences within the figures, specifically the main treatment effects (CO₂ and *Ulva*).**

*Reviewer #2:*

*"The ability of macroalgae to mitigate the negative effects of ocean acidification on four species of North Atlantic bivalve" This paper evaluates the effect of the presence of the macroalga Ulva rigida on the growth of four North Atlantic bivalve species, Mercenaria mercenaria, Crassostrea virginica, Argopecten irradians and Mytilus edulis. The authors have used small and larger sizes of three out of four species, specifically the three obtained from hatcheries. The pCO2 levels the bivalves are exposed to are high, but conceivable for estuarine systems. The authors claim that "saturation states for calcium carbonate (Ω) were significantly higher in the presence of Ulva under both ambient and elevated CO2 delivery rates (p<0.05)", and that "alkalinity was increased by the presence of Ulva". This might be statistically significant, but as alkalinity actually decreases (or is similar) in some treatments (small Mercenaria, large Mercenaria control pH, small Crassostrea, large Crassostrea low pH) it would be interesting to see the relationship between these parameters and growth directly and visually.*

**We agree with the reviewer's assessment. We agree with the reviewer's suggestion that plots of saturation states against the growth would be important. As suggested by the reviewer here, for this revision, we have made plots showing growth metrics of each species by treatment, with aragonite saturation state of each treatment on the x-axis. We have placed the resulting statistics in tables as supplements to the manuscript (new Table S10), with references throughout the manuscript. To summarize these findings, there were strong positive and significant ($p<0.05$) correlations between shell length-based growth and saturation states of aragonite and calcite for all species and size classes, save for the single *Mytilus edulis* experiment. In at least half of the experiments, there was a strong positive correlation and significant ($p<0.05$) correlation between tissue and shell weight-based growth and the saturation states of calcite and aragonite, with several additional results approaching significance ($p<0.07$). Regarding alkalinity, we note that it**

**is affected by many processes and while nitrate uptake will increase alkalinity, other processes may decrease it and that prior research has definitively demonstrated that saturation states for calcium carbonate are the key factor dictating the effects of acidification on bivalves.**

5   *In treatments with Ulva additions, one would expect the variability in pH to be higher due to respiratory activity and production. However, the average pH is higher but the variability in pH seems similar to treatments without Ulva. In fact, I would expect the algal-addition treatments to have a fluctuating pH and the control treatments to be stable, which could arguably have caused the differences. However the authors do not discuss this and the tables do not show these differences in*
10  *variability of pH. Was the pH fluctuating on a day-night scale in the Ulva treatments? Or was the gas flowrate so high this was not discernable, and what causes the variability in the control treatments?*

**The reviewer is correct that the treatments with *Ulva* had more variability in pH but did, on average, have higher pH levels.  For this revision, we have added plots (Figures S2-S3) showing**
15  **the changes in pH over time for the *Ulva* treatments to demonstrate that there is variability but that the pH rose in these treatments after each water change, likely due to the uptake of nitrate and the assimilation of $CO_2$.**

*The nutrient and algae addition to the vessels might cause different nutrient concentrations in the*
20  *treatments, with Ulva taking up nutrients while they remain suspended in the control vessels, which could have influenced results.*

**We agree with the reviewer that additions of nutrients and algae might cause different nutrient concentrations within treatments, and that *Ulva* may alter nutrient concentrations. One sign of**
25  **different nutrient levels effecting the bivalves would be via higher levels of phytoplankton in vessels without Ulva which could yield more growth in the bialves. However, for this revision we added the enumerations of phytoplankton concentrations, which were found to be similar and not significantly different between treatments across all experiments.**

30  **Importantly, we also note that any differences in nutrients among the vessels would occur in an ecosystem setting as well with more nitrate assimilation and removal and thus an increase in alkalinity in times and places where there is more *Ulva*.  Hence, any differences on this front would be realistic in an ecosystem setting.**

35  *It is unclear what time of the year the experiments have been done (presumable summer due to hatchery times), and how the results might vary in other seasons (i.e. when Ulva is not productive).*

**The reviewer's presumption is correct as the experiments occurred throughout summer 2017, which is the peak growing season for bivalves and *Ulva*. We targeted this season specifically for**
40  **that reason, although it should be noted that within the collection site of *Ulva*, the macroalgae appear during the early days of spring, and persists into the end of the fall months, which is beyond the time that our experiments were concluded.  During period when *Ulva* grows more**

**slowly (spring and fall) it would be expected that its growth would be slower and its ability to mitigate acidification would be lower.**

*The various sizes and the amount of different species of bivalves used in this study make it an interesting read, even though it is not entirely clear what causes the beneficial effect of the presence of Ulva (its effect on the carbonate chemistry, nutrient concentration or something else).*

**We agree with the reviewer that the exact cause of the increased bivalve growth in the presence of *Ulva* is not entirely known. We believe the new regression analyses we have included that depicted the significant linear relationships between calcium carbonate saturation states and bivalve growth across all treatments in six of seven experiments makes the carbonate chemistry angle more convincing. Similarly, our newly included daily pH data that shows pH levels were consistently higher within 24 h of each water change in treatments with *Ulva*. Further, our inclusion of phytoplankton density data showing there are no differences among treatments indicate this was not a driver of the findings.**

*Specific comments: Methods P.3, line 9 "light intensity (∼200 μmol photons m-2 s-1)", how does this compare to ambient conditions?*

**Light intensity used in all experiments was set to mimic ambient light intensity where *Ulva* grows in near shore regions. We have added this text to the manuscript to specify this.**

*P.3, line 23: Isochyrsis should be Isochrysis*

**We have made the suggested change.**

*P.4, line 17: "some estuarine environments" – representable for the environments of the study organisms and their origin?*

**Yes. For example, Wallace et al. (2014) observed $pCO_2$ concentrations exceeding 2,000 μatm in Jamaica Bay, NY, USA, which hosts the bivalve and macroalgae species used in the present study.**

*P.4, line 32-33: "Well-pigmented, circular sections of Ulva (∼3.5 cm and ∼7 cm for experiments in small containers and large vessels, respectively". These small containers where 1L, while the large vessels had a volume of 8L. The biomass of Ulva however, is 2x as large for the larger volume, which does not respect the ratio biomass/water volume. The authors state that the weight was consistent with the benthic coverage in Shinnecock Bay, would that mean that the 8L vessels had 2x the diameter of the small containers and would water volume not be more important than surface in this case? Or was there more than 1 disk per container (p.5., line 23 states "disks")? This section is a bit unclear.*

**The amount of *Ulva* used was based on tissue weight, not tissue surface area, and the amount of *Ulva* added to 1 L and 8 L containers was consistent with the benthic coverage of *Ulva* in**

**Shinnecock Bay based on several years of benthic trawl data as well as other estuarine regions (Liu et al., 2015; Sfriso et al., 2001). This point is specified in the Methods on P5, L4-8. Considering the 2-dimensional nature in which interactions of the bivalves and the macroalgae would occur, it made more sense to base the weight of macroalgae used on the surface area of the container, and not necessarily the volume.**

*P.5, line 16-17: "with discrete and continuous measurements of pH, dissolved oxygen, and temperature", which measurements were discrete and which continuous?*

**We measured pH and temperature discretely and dissolved oxygen continuously. We have changed the text to specify this difference.**

*Results P.6, lines 19-20: "For the larger-sized cohort of M. mercenaria ($5.00 \pm 0.41$ mm), $\Omega calcite$ and $\Omega aragonite$ were significantly higher in treatments containing Ulva and significantly lower in high CO2 treatments" Throughout the manuscript's result section this way of describing the differences between high CO2 / Ulva treatments is confusing. In the highCO2+Ulva treatment the $\Omega calcite$ is actually lower than the control-Ulva treatment (as expected), however from the text it appears at a first glance that all Ulva containing treatments are higher, the sentences might be clarified to prevent confusion.*

**We intended to specify that $\Omega_{calcite}$ and $\Omega_{aragonite}$, although significantly lower under elevated $CO_2$ concentrations in general, were significantly higher in the presence of *Ulva* in both ambient and elevated $CO_2$ treatments. We agree with the reviewer that the sentence structure used throughout the manuscript may cause confusion and have changed the text to separate any significant differences in $\Omega_{calcite}$ and $\Omega_{aragonite}$, be it under elevated $CO_2$ conditions, or in the presence of *Ulva*. We have also included references to the respective figures that show $\Omega_{calcite}$ and $\Omega_{aragonite}$, which would make it clear that $\Omega_{calcite}$ and $\Omega_{aragonite}$ are lower under elevated $CO_2$, but higher in the presence of *Ulva* in both ambient and elevated $CO_2$ treatments.**

*Discussion Could the fact that Mytilus seems less sensitive to addition of Ulva be related to the more "natural" (no hatchery) origin of the juveniles and their exposure to environmental fluctuations vs. the more stable hatchery conditions?*

**This is a good point raised by the reviewer. The area within Shinnecock Bay where *Mytilus* were collected is well-flushed and not prone to significant decreases in dissolved oxygen or pH or increases in $pCO_2$. In addition, *Mercenaria* and *Argopecten* within the hatchery at Stony Brook University in Southampton are exposed to similar environmental conditions that are found in the collection sites in Shinnecock Bay from which these original broodstock came. We have clarified the recent origin of the broodstock used in experiments in the methods.**

*If the presence of algae buffered the carbonate chemistry (p.9, line 23) and this is the mechanism for enhanced growth, this should be visible when $\Omega calcite/aragonite$ is plotted vs. growth. However, the saturation state with Ulva is still considerably below 1 in the highCO2 treatments and the SD is high.*

**This was an excellent suggestion by the reviewer and for this revision, we have included regression of $\Omega_{calcite}$ and $\Omega_{aragonite}$ vs. growth which in nearly all cases provided significant correlations.  While the $\Omega$ is below 1 in many cases, prior studied have shown bivalves can grow, albeit slower, under such conditions (Talmage and Gobler 2010, 2011).**

*Did the authors measure nutrients at the end of the incubations? It would be interesting to explore their theory that through Ulva presence "
[revised manuscript text omitted]

---

## Referee Report (RR2)

The manuscript entitled "The ability of macroalgae to mitigate the negative effects of ocean acidification on four species of North Atlantic bivalves" reports on a large and impressive experimental study to assess the effects of elevated $p$CO2 and the occurrence of green macroalgae on the growth and survival of calcifying bivalves. The main result is that macroalgae may mitigate the deleterious effects of acidification on bivalves. In the context of increasing human pressure on ecosystems, there is a definite need/interest in understanding what are the consequences of acidification on marine ecosystems. And it is particularly relevant in situations where communities are composed of species that usually play a very substantial role in ecosystem functioning, besides being of commercial interest. This study adds on current research on the effects of acidification on the functioning of marine ecosystems. The authors did pay attention to all the comments made by the reviewers and made significant changes to improve the ms. New analyses have been undertaken that strengthen the results and the discussion has been amended accordingly. The topic clearly meets the criteria laid down for publication in Biogeosciences. The manuscript is well written and could be accepted for publication as is. I think the manuscript can be a very valuable addition to Biogeosciences.

---

## Author Response (AR2)

RESPONSE TO REVIEWER COMMENTS
*Reviewer comments in italics*; **author responses to bold**

**Reviewer #1:**

*The authors have addressed most of my concerns, but it seems the main issue is still not resolved. The data do not show that the increase in growth of bivalves in treatments with Ulva is due to changes in saturation state.*

**We thank the reviewer for their feedback. We have made numerous changes to the manuscript to address the main concerns of the reviewer, which are detailed below.**

*The authors provided new regression analyses showing that growth is correlated to saturation state. The figure is not shown, but after reproducing the data, it appears these regressions pool all the data within an experiment, without addressing the specific effect of Ulva.*

**There was no figure associated with the regression analyses as we instead presented them as supplementary tables. The reviewer is correct that all data within an experiment were pooled when creating the regressions. As far as addressing the specific effect of *Ulva*, our new Supplementary Figures S2 and S3 demonstrate that the presence of *Ulva* increased daily pH consistently for the duration of experiments in ambient and elevated CO$_2$ treatments. We note that across all seven experiments the maximum difference in saturation states of aragonite based on the maximal pH difference based on daily measurements was ~0.1 and ~0.5 units in elevated and ambient CO$_2$ treatments, respectively. Importantly, below we describe while the actual differences were likely even larger than this.**

*This analysis does not contribute anything new and does not address my original comment, which was: "...a statistically significant difference in a carbonate chemistry parameter across treatments does not mean that it is biologically relevant. The authors do not discuss if the magnitude of change in growth is realistic for a 0.04 change in saturation state."*

*To clarify my point, I extracted data from all their figures and plotted shell weight growth (mg/day) by aragonite saturation state (example for Fig. 1 shown below, Mercenaria). The CO2/Ulva treatment only increased saturation state by a very small amount (0.03). Assuming a linear correlation between growth and saturation state (as identified by the authors), in the absence of Ulva, one would need a saturation state >1.5 to achieve the same increase in shell weight growth that was observed in the CO2/Ulva treatment (red arrows).*

*This same pattern is observed for the data in Fig. 2 (Mercenaria), Fig. 5, and Fig. 6 (Agropectin). This pattern does not hold for Fig. 3 and 4 (Crassostrea). The fact that growth increased in Fig. 7 (Mytilus) in Ulva treatments, despite the unexpected effect of saturation state (i.e., pCO2 effect), again suggests*

*that Ulva provides a positive effect on growth, but this affect cannot be explained by changes in aragonite saturation state.*

**We agree with that reviewer that a higher saturation state would be required to linearly correlate with the increase in bivalve growth in the presence of *Ulva*, and that bivalve growth is disproportionate to the reported changes in saturation state. We are very highly appreciative of the reviewer's persistence on this point and the figure he/she made, as it has allowed us to even more closely and deeply examine our data, hypotheses, and analyses. Through that process, we considered what could account for the discrepancy between our measured saturation states and growth and considered the potential diurnal changes in pH relative to the timing of our pH and DIC measurements. For this revision, we have added a figure to our discussion that shows the change in pH in ambient and elevated $CO_2$ treatments in the presence of *Ulva* over 24 hours. The data shows a diurnal pattern of pH, with levels rising during the day due to photosynthesis, and declining at night due to respiration. Not fully expecting or appreciating this pattern, discrete pH measurements were made during this study in the late morning every day (9:00-11:00 AM) when pH values were typically only slightly above their daily minimum; pH peaks values in the evening were 0.25-0.30 units higher than those recorded in the morning. Unfortunately, we did not have a continuous pH sensor for all vessels during all experiments, but rather used one at a time in singular vessels during occasional experiments and only downloaded and assessed the data trends after experiments had been completed.**

**In light of these trends, it seems likely that reported $\Omega_{aragonite}$ and $\Omega_{calcite}$ values represent underestimates of the true mean conditions that bivalves were exposed to in treatments with *Ulva* in ambient and elevated $CO_2$ conditions over the course of experiments. We calculated that saturation states of calcite and aragonite would have been higher by 0.60 and 0.40 units in the elevated $CO_2$ treatment had they been taken towards the end of the day. Our DIC samples were also collected in late morning and we expect that those values would also be higher had they been collected mid-day or in the afternoon. This sampling strategy would also have contributed to our reported $\Omega_{aragonite}$ and $\Omega_{calcite}$ values being underestimates. For our revision, we have outlined and highlighted this important information. We have also added an additional paragraph to the Discussion that highlights alternative factors to consider, such as food availability and dissolved oxygen levels.**

*My conclusion from this data is that the presence of Ulva in a closed experimental system provided an unknown positive affect on shell growth for 3 of the 4 tested species. The change is growth is much greater than what can be expected from the observed Ulva-induced changes in aragonite saturation state alone.*

**We agree with the reviewer that increases in aragonite saturation states in the presence of *Ulva* may not be the sole factor that increased bivalve growth rates while also emphasizing that these values are likely underestimates. We have included an additional paragraph in the Discussion that details the factors driving this underestimation as well as several alternative hypotheses to**

**explain why increased bivalve growth in the presence of *Ulva* was observed including food availability and dissolved oxygen. Regarding the discrete pH underestimating the actual increases in calcium carbonate saturation states, we have included an additional figure in the supplement that shows continuous pH measurements over 24 hours, which captures the trend in pH driven by the daytime and nighttime photosynthesis and respiration of *Ulva*, respectively, in ambient and elevated $CO_2$ conditions.**

*The emphasis on saturation state throughout the Results and Discussion ignores this aspect of the data, leading to the incorrect conclusion that the increased saturation state by Ulva caused an increase in shell growth. Saturation state alone does not explain the observed biological response. Therefore, I suggested that the authors provide alternative hypotheses.*

**We agree with the author that the changes we observed in carbonate chemistry may be disproportionate to the increases in bivalve growth in the presence in *Ulva*. As such, we have added a new paragraph that highlights other possible factors, such as the underestimation of $\Omega_{aragonite}$ and $\Omega_{calcite}$ values. We have also included an additional figure to the supplement which details the change in pH and, by extension, carbonate chemistry in the presence of *Ulva* over 24 hours in ambient and elevated $CO_2$ conditions.**

*Reviewer #2:*

*The manuscript entitled "The ability of macroalgae to mitigate the negative effects of ocean acidification on four species of North Atlantic bivalves" reports on a large and impressive experimental study to assess the effects of elevated pCO2 and the occurrence of green macroalgae on the growth and survival of calcifying bivalves. The main result is that macroalgae may mitigate the deleterious effects of acidification on bivalves. In the context of increasing human pressure on ecosystems, there is a definite need/interest in understanding what are the consequences of acidification on marine ecosystems. And it is particularly relevant in situations where communities are composed of species that usually play a very substantial role in ecosystem functioning, besides being of commercial interest. This study adds on current*
*research on the effects of acidification on the functioning of marine ecosystems. The authors did pay attention to all the comments made by the reviewers and made significant changes to improve the ms. New analyses have been undertaken that strengthen the results and the discussion has been amended accordingly. The topic clearly meets the criteria laid down for publication in Biogeosciences. The manuscript is well written and could be accepted for publication as is. I think the manuscript can be a very valuable addition to Biogeosciences.*

**We thank the reviewer for their recommendation.**

[revised manuscript text omitted]